# Frequency Matching in Spiking Neural Networks for mmWave Sensing

**Zhenyu Liao** [* 1]  **Di Yu** [* 1]  **Changze Lv** [2]  **Wentao Tong** [1]  **Linshan Jiang** [3]  **Sijie Ji** [4]  **Xin Du** [5]  **Hailiang Zhao** [5]
**Xiaoqing Zheng** [2]  **Shuiguang Deng** [1]

## Abstract

Millimeter-wave (mmWave) sensing enables privacy-preserving, always-on edge perception, but its measurements are often sparse, temporally irregular, and corrupted by high-frequency noise. Existing mmWave pipelines predominantly rely on artificial neural networks (ANNs), which achieve robustness through extensive preprocessing or deep architectures, thereby limiting their efficiency on edge devices. In this work, we study spiking neural networks (SNNs) for mmWave sensing from a mechanism–data alignment perspective. By leveraging the low-pass filtering behavior of leaky integrate-and-fire (LIF) dynamics, we analyze how their implicit temporal filtering interacts with the frequency structure of mmWave signals. Our analysis shows that when discriminative information resides in low-to-mid frequencies, LIF dynamics can inherently suppress high-frequency noise, clarifying when and why SNNs outperform ANNs. Based on this insight, we derive a principled criterion for configuring the membrane decay factor by matching the effective bandwidth of LIF dynamics to the data's discriminative spectral content. Experimental results across four widely used mmWave datasets validate the proposed frequency-matching hypothesis, yielding an average test-accuracy improvement of 6.22% and a 3.64× reduction in theoretical energy consumption relative to ANN baselines, under a unified evaluation protocol.

---

[*]Equal contribution  [1]College of Computer Science and Technology, Zhejiang University, Hangzhou, China [2]School of Computer Science, Fudan University, Shanghai, China [3]Department of Computer Science and Engineering, Southern University of Science and Technology, Shenzhen, China [4]California Institute of Technology, Pasadena, United States [5]School of Software Technology, Zhejiang University, Ningbo, China. Correspondence to: Xin Du <xindu@zju.edu.cn>, Linshan Jiang <jiangls@sustech.edu.cn>, Shuiguang Deng <dengsg@zju.edu.cn>.

*Proceedings of the 43$^{rd}$ International Conference on Machine Learning*, Seoul, South Korea. PMLR 306, 2026. Copyright 2026 by the author(s).

## 1. Introduction

With the rapid expansion of the Internet of Things (IoT) (Siam et al., 2025), wireless sensing has emerged as a critical enabler for applications such as human tracking (Zhou et al., 2025; Liu et al., 2024), safety monitoring (Chang et al., 2024; Zolfagharian et al., 2024), and autonomous driving (Guo et al., 2025; Wang et al., 2024). Among existing sensing modalities, millimeter-wave (mmWave) sensing (Li et al., 2025; Choi et al., 2025) offers several distinctive advantages: robustness to illumination variations, strong penetration capability, and the ability to capture rich geometric and kinematic information (e.g., range, velocity, and angle). In addition, mmWave sensing provides stronger privacy guarantees than vision-based systems, making it particularly attractive for reliable, information-dense perception in diverse environments and for always-on deployment on resource-constrained edge devices (Gong et al., 2025).

Despite these advantages, mmWave measurements are inherently sparse, temporally irregular, and often corrupted by multipath-induced artifacts and high-frequency noise, such as phase jitter and interference (Xing et al., 2024; Cui et al., 2023). These characteristics pose significant challenges to stable and accurate perception.

Existing mmWave sensing pipelines predominantly rely on artificial neural networks (ANNs), achieving performance gains either through extensive handcrafted preprocessing or by adopting deeper and more complex architectures (Duan et al., 2025; Liu et al., 2025). Both strategies substantially increase computational cost and inference latency, thereby conflicting with the stringent efficiency constraints of edge platforms. This tension underscores the need for learning paradigms that better balance accuracy and efficiency by aligning with the intrinsic properties of mmWave signals.

This imperative for efficiency motivates the adoption of spiking neural networks (SNNs) (Ghosh-Dastidar & Adeli, 2009) as a promising alternative to ANNs. By suppressing redundant activations and replacing multiply–accumulate (MAC) operations with lightweight spike-based computations, SNNs inherently favor energy-efficient processing (Wei et al., 2025). Indeed, SNNs have already been deployed in several edge-sensing benchmarks and demon-

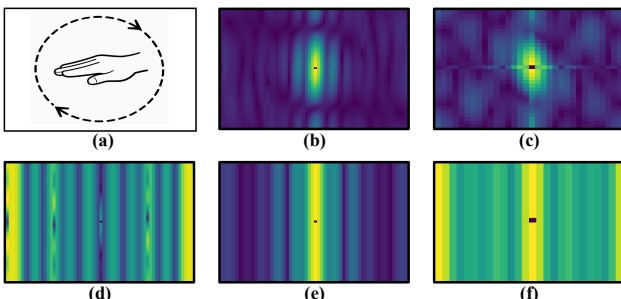

*Figure 1.* (a) An mmWave input and (d) its corresponding spatial spectrum. (b,c) Feature spectra after the first and second convolutional layers of the ANN, and (e,f) feature spectra after the first and second spiking convolutional layers of the SNN, illustrate how each model progressively reshapes spectral energy across layers.

strate accuracy comparable to that of ANNs (Yu et al., 2025; Lv et al., 2024b). These properties position SNNs as a viable pathway toward energy-efficient, always-on mmWave inference on resource-limited edge devices (Safa et al., 2021).

While existing studies on SNN-based mmWave sensing have demonstrated the feasibility and empirical benefits of deploying SNNs in this domain, most work emphasizes energy efficiency or latency advantages (Zheng et al., 2024; Arsalan et al., 2023), or reports performance gains achieved through task-specific tuning (Arsalan et al., 2022b; Safa et al., 2021). As a result, it remains unclear when and why SNNs are advantageous for mmWave sensing, and how their neuronal dynamics should be configured to effectively exploit signal characteristics. To address this gap, we study mmWave sensing through the lens of mechanism–data alignment, explicitly characterizing the correspondence between neuronal dynamics in SNNs and the spectral and temporal structure of mmWave measurements.

Specifically, we analyze mmWave datasets in the frequency domain using the Discrete Fourier Transform (DFT) (Proakis & Manolakis, 2007), as illustrated in Figure 1. Based on this representation, we introduce the notion of a discriminative spectrum (Fridovich-Keil et al., 2022) to characterize the distribution of task-relevant information across frequency bands.

Motivated by these data characteristics, we leverage the equivalence between the discrete-time dynamics of the leaky integrate-and-fire (LIF) neuron (Gerstner et al., 2014) and a first-order infinite-impulse-response (IIR) low-pass filter (Fang et al., 2025). This connection enables an explicit characterization of the LIF neuron's spectral response (He et al., 2021) and effective bandwidth (Chen et al., 2024), providing a mechanistic basis for analyzing how spiking dynamics interact with mmWave signal spectra.

Our analysis indicates that when discriminative information is concentrated in the *low-to-mid* frequency range and

*high-frequency* components are predominantly noisy, SNNs can naturally suppress noise while preserving salient cues through their neuronal dynamics. Conventional ANNs, in contrast, lack such an intrinsic temporal-filtering inductive bias and therefore exhibit reduced robustness under noisy mmWave conditions. To quantify this mechanism–data correspondence, we define an average spectral matching score that measures the alignment between the LIF-induced filtering bias and the data's discriminative spectral structure. Building on this insight, we further derive a principled criterion for configuring the membrane decay factor: the effective bandwidth of LIF dynamics should be matched to the discriminative frequency content of the mmWave signals, balancing noise attenuation and information preservation.

We evaluate the proposed frequency-matching hypothesis on *four* widely used mmWave sensing datasets[1]. Under a unified training and evaluation protocol, a LeNet-style SNN consistently outperforms a diverse set of commonly used ANN baselines for mmWave sensing (Yang et al., 2023), achieving an average test-accuracy improvement of 6.22% while reducing theoretical energy consumption by approximately 3.64×. Additional ablation studies and spectral analyses further corroborate the proposed decay-factor selection rule, demonstrating that aligning the effective bandwidth of LIF neuronal dynamics with the discriminative spectral band of mmWave signals leads to consistent and robust performance gains. Overall, these results suggest that the frequency-matching principle provides a unifying explanation for the empirical advantages of SNNs in noisy mmWave sensing scenarios and offers practical guidance for spiking model design under real-world signal conditions.

Our main contributions are summarized as follows:

- We provide a mechanistic analysis that explains when and why the temporal dynamics of spiking neurons are well matched to the spectral structure of mmWave signals.
- Building on this analysis, we propose a principled criterion for setting the membrane decay factor, aligning spiking neuron dynamics with the discriminative frequency bands of mmWave signals.
- Extensive experiments on four widely used mmWave datasets validate the frequency-matching hypothesis, showing that SNNs consistently achieve higher accuracy and greater energy efficiency than ANN baselines.

## 2. Preliminaries

**Spiking Neural Networks.** SNNs (Ghosh-Dastidar & Adeli, 2009) process information over discrete timesteps using event-driven binary spikes and are widely studied as energy-efficient alternatives to conventional ANNs (Sze et al., 2017). Unlike ANNs, SNNs are explicitly gov-

---

[1] Code is available at: https://github.com/yudi-mars/Soul

erned by neuronal dynamics that integrate inputs over time, thereby introducing structured temporal inductive biases. Among commonly used spiking neuron models, the LIF neuron (Gerstner et al., 2014) is the de facto standard due to its simplicity and its ability to capture essential temporal integration behavior. In discrete time, the dynamics of a LIF neuron are given by $u_{t+1} = \beta\, u_t + (1 - \beta)\, I_t - v_{\text{th}} \cdot O_t$, where $u_t$ denotes the membrane potential at timestep $t$, $I_t$ is the synaptic input current, and $\beta \in [0, 1)$ is the membrane decay factor controlling temporal integration. The spike output is defined as $O_t = \Theta[u_t \geq v_{\text{th}}]$, where $v_{\text{th}}$ is the firing threshold and $\Theta[\cdot]$ denotes the Heaviside step function (Yu et al., 2025). For linearized analysis, ignoring the reset term yields a first-order IIR low-pass filter (Fang et al., 2025). Consequently, LIF dynamics inherently preserve low-frequency signal components while attenuating high-frequency fluctuations, thereby inducing an implicit spectral bias that favors data whose discriminative information lies in the low-to-mid-frequency range.

**Millimeter-Wave Sensing.** mmWave radar is widely adopted for edge sensing due to its high temporal resolution, robustness to illumination variations, and strong penetration capability, making it well-suited for resource-constrained deployments. In a typical sensing pipeline, the radar transmits mmWave signals and records target reflections, producing time-varying echo sequences. Following standard preprocessing procedures (Palipana et al., 2021), these measurements are organized into a supervised dataset $\mathcal{D} = \{(\mathbf{X}_i, y_i)\}_{i=1}^M$, where each sample $\mathbf{X}_i \in \mathbb{R}^{L \times C \times H \times W}$ consists of $L$ consecutive frames with $C$ channels and sensing-map resolution $H \times W$, and $y_i \in \{1, \ldots, \mathcal{C}\}$ is the corresponding class label. In practice, mmWave measurements are often corrupted by multipath-induced artifacts and high-frequency noise, which can adversely affect recognition stability and accuracy (Cui et al., 2023). As LIF neurons impose an intrinsic low-pass filtering bias (Fang et al., 2025), SNNs may naturally suppress such perturbations while preserving task-relevant discriminative structure.

# 3. Methods

## 3.1. Problem Statement

To understand when and why spiking dynamics benefit mmWave sensing, we take a frequency-domain view that connects the spectral properties of mmWave signals with the temporal filtering behavior of spiking neurons. From this perspective, mmWave recognition amounts to selectively preserving task-relevant frequency components while suppressing pervasive high-frequency noise.

**Definition 3.1** (Frequency-Selective mmWave Recognition). Given an mmWave sample $\mathbf{X}_i \in \mathbb{R}^{L \times C \times H \times W}$ with $L$ frames, let $\mathcal{F}_t$ denote the DFT applied along the temporal axis, and define the temporal spectrum $\hat{\mathbf{X}}(\omega) \triangleq \mathcal{F}_t(\mathbf{X}_i)(\omega)$

indexed by discrete frequency $\omega$. We model:

$$\hat{\mathbf{X}}(\omega) = \hat{\mathbf{S}}(\omega) + \hat{\mathbf{N}}(\omega) \tag{1}$$

where $\hat{\mathbf{S}}(\omega)$ and $\hat{\mathbf{N}}(\omega)$ denote task-relevant signal and noise components, respectively. mmWave recognition aims to extract task-relevant components from a discriminative band $\Omega^\star$ and predict the label via

$$\hat{y} = g\Big(\{\hat{\mathbf{X}}(\omega)\}_{\omega \in \Omega^\star}\Big) \tag{2}$$

where $g(\cdot)$ denotes the classifier applied to the selected spectral components, and $\Omega^\star$ denotes the task-relevant discriminative frequency subset (i.e., the band in which class-separating information concentrates).

To ground $\Omega^\star$ in the data, we next characterize mmWave sequences in the frequency domain and identify where discriminative information concentrates.

## 3.2. Frequency Analysis of mmWave Signals

mmWave measurements are discrete-time signals whose discriminative information is distributed across frequency components. Following prior work (Zhao et al., 2023; Brighente et al., 2020), we apply the DFT to characterize their spectral structure and enable frequency-domain analysis.

For each sample $\mathbf{X}_i \in \mathcal{D}$, we compute discrete spectra for each frame $\mathbf{X}_i[l]$, $l \in \{0, \ldots, L-1\}$ on a DFT grid and define the one-sided frequency grid as:

$$\Omega_L \triangleq \left\{ \omega_k = \frac{2\pi k}{L} \,\Big|\, k = 0, 1, \ldots, K \right\}, \ K \triangleq \left\lfloor \frac{L}{2} \right\rfloor \tag{3}$$

where $\omega_k$ is the $k$-th DFT angular frequency and $K$ is the largest one-sided index.

To obtain a dataset-level, layout-agnostic diagnostic, we compute a discriminative index (DI) on a scalar temporal proxy by averaging each frame over non-temporal axes, and evaluate it on the common one-sided grid $\Omega_L$ (Fukunaga, 2013). Concretely, for each sample $\mathbf{X}_i$, we construct a scalar temporal sequence $\mathbf{s}_i \in \mathbb{R}^L$ by averaging each frame $\mathbf{X}_i[l]$ over the non-temporal dimensions $(C, H, W)$, yielding $\mathbf{s}_i[l] \in \mathbb{R}$, and we apply a fixed, sample-wise de-meaning preprocessing rule to obtain $\bar{\mathbf{s}}_i[l]$. We then compute the one-sided DFT of $\bar{\mathbf{s}}_i$ on $\Omega_L$, yielding coefficients $\tilde{\mathbf{s}}_i[k] \in \mathbb{C}$; we discard phase and use the amplitude spectrum $A_i[k] \triangleq |\tilde{\mathbf{s}}_i[k]| \in \mathbb{R}_{\geq 0}$ as the per-frequency feature.

For each frequency bin $k$, we estimate the per-class mean $\mu_c[k]$ and the unbiased intra-class variance $\text{Var}_c[k]$ of the scalar feature $A_i[k]$. Based on these statistics, we construct Fisher-style inter-class scatter $\text{S}_\text{B}$ and intra-class scatter

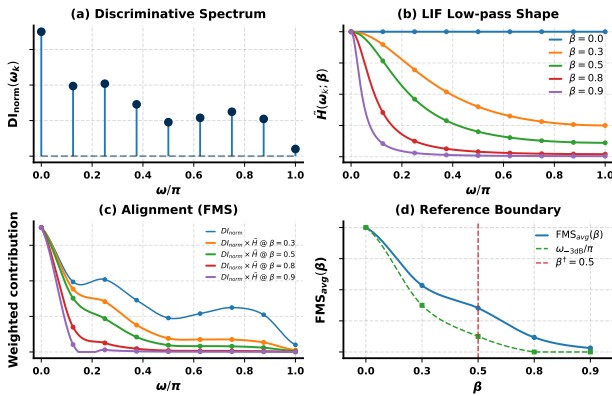

*Figure 2.* (a) Normalized discriminative spectrum of a sample in AOPHand (Zafar, 2023), showing mass concentrated in low–mid bands with residual high-frequency cues. (b) LIF low-pass template $\tilde{H}(\cdot;\beta)$ under monotone bandwidth control; larger $\beta$ increases low-pass strength. (c) Mechanism–data alignment quantified by FMSavg($\beta$); decreases with $\beta$ due to reduced retention. (d) Reference boundary $\beta^{\dagger}$ (red dashed line) derived from $\text{FMS}_{\text{avg}}(\beta)$, marking onset of over-low-pass behavior.

$S_W$ (Murphy, 2022), denoted as:

$$S_B[k] = \sum_{c=1}^{\mathcal{C}} \pi_c \big( \mu_c[k] - \bar{\mu}[k] \big)^2 \qquad (4)$$

$$S_W[k] = \sum_{c=1}^{\mathcal{C}} \pi_c \, \text{Var}_c[k] \qquad (5)$$

where $\bar{\mu}[k] = \sum_c \pi_c \mu_c[k]$ denotes the global mean at frequency bin $k$, and $\pi_c$ is the empirical class prior estimated from the training set. We then define the per-bin DI as:

$$\text{DI}(\omega_k) := \frac{S_B[k]}{S_W[k] + \varepsilon} \qquad (6)$$

where a small constant $\varepsilon > 0$ is added for numerical stability. Larger $\text{DI}(\omega_k)$ implies stronger task-relevant cues at $\omega_k$ and thus has greater potential contribution to higher performance (Chang, 2023). Finally, we normalize $\text{DI}(\omega_k)$ over $\Omega_L$ to obtain a one-sided distribution:

$$\text{DI}_{\text{norm}}(\omega_k) = \frac{\text{DI}(\omega_k)}{\sum_{\omega_k \in \Omega_L} \text{DI}(\omega_k)} \qquad (7)$$

such that $\text{DI}_{\text{norm}}$ forms a probability mass function on $\Omega_L$ and is invariant to global rescaling. In Figure 2(a), a representative example of the normalized discriminative information spectrum $\text{DI}_{\text{norm}}$ is illustrated. The complete derivation of this section is provided in *Appendix A*.

### 3.3. Spectral Response Study of LIF Dynamics

Figure 2(a) shows that although mmWave measurements are often dominated by high-frequency noise, they can still contain discriminative information at higher frequencies. Accordingly, prior work (Han et al., 2024; Lee et al., 2019) typically applies an explicit front-end low-pass filter for denoising. Nevertheless, such preprocessing performs hard thresholding in the frequency domain, indiscriminately suppressing both noise and informative high-frequency components, thereby degrading recognition performance.

In contrast, the LIF dynamics in SNNs impose an intrinsic low-pass bias (Fang et al., 2025) that selectively attenuates high-frequency perturbations without uniformly eliminating the entire high-frequency band, thereby better preserving informative high-frequency structure. This contrast motivates our mechanism–data alignment analysis of SNNs for mmWave sensing. As a first-order IIR filter, the LIF neuron admits the frequency response $H(\omega_k;\beta) = (1 - \beta e^{-j\omega_k})^{-1}$ (Oppenheim & Schafer, 2010; Proakis & Manolakis, 2007), where $j$ is imaginary unit. To factor out the overall amplitude, we define the Direct Current (DC)-normalized power template (Pfister, 2017) on the one-sided grid $\Omega_L \in [0, \pi]$ as:

$$\begin{aligned} \tilde{H}(\omega_k;\beta) &\triangleq \frac{|H(\omega_k;\beta)|^2}{|H(0;\beta)|^2} \\ &= \frac{(1-\beta)^2}{(1-\beta)^2 + 2\beta(1-\cos\omega_k)} \end{aligned} \qquad (8)$$

Building on the above analysis, we formalize the following lemma and definition characterizing LIF's low-pass spectral property and monotone bandwidth control (see Figure 2(b)).

**Lemma 3.2** (Low-pass shape and monotone bandwidth control of LIF)**.** *Let* $\beta \in [0, 1)$ *and consider* $\tilde{H}(\omega_k;\beta)$ *over* $\omega_k \in [0, \pi]$. *Then, for all* $\omega_k \in [0, \pi]$:

*(i)* ***Boundedness and DC normalization:*** $\tilde{H}(\omega_k;\beta) \in (0, 1]$ *and* $\tilde{H}(0;\beta) = 1$;

*(ii)* ***Monotonicity in frequency:*** $\tilde{H}(\omega_k;\beta)$ *is non-increasing in* $\omega_k$;

*(iii)* ***Monotonicity in leak:*** *for any fixed* $\omega_k \in (0, \pi]$, $\tilde{H}(\omega_k;\beta)$ *is non-increasing in* $\beta$.

*Consequently, increasing* $\beta$ *suppresses all non-DC frequency components monotonically and narrows the effective passband (see* ***Appendix B*** *for a formal proof).*

**Definition 3.3** (Half-power cutoff and effective bandwidth)**.** Fix $\beta \in [0, 1)$ and let $\tilde{H}(\omega_k;\beta)$ denote the DC-normalized power template. The **half-power** ($-3\,\text{dB}$ (Proakis & Manolakis, 2007)) **cutoff** $\omega_c \in (0, \pi]$ is $\tilde{H}(\omega_c;\beta) = \frac{1}{2}$, and the one-sided **effective bandwidth** is $B_{\text{eff}}(\beta) \triangleq \omega_c$.

Lemma 3.2 and Definition 3.3 establish that LIF induces a DC-normalized monotone low-pass bias, with $\beta$ acting as a principled inverse-bandwidth control via the half-power cutoff $B_{\text{eff}}(\beta)$ (see ***Appendix C*** for the complete derivation). Next, we connect this controllable spectral retention to the

**Algorithm 1** Criterion of the reference boundary $\beta^\dagger$

---

1: **Input:** dataset $\mathcal{D} = \{(X_i, y_i)\}_{i=1}^M$, training split $\mathcal{D}_{tr} \in \mathcal{D}$ ; candidate set $\mathcal{B} = \{\beta_r\}_{r=1}^R$ (sorted ascending).
2: Compute $\mathrm{DI}_{\mathrm{norm}}(\omega_k)$ on $\mathcal{D}_{tr}$.  // `Eq.(3)-(7)`
3: **for** $r = 1$ to $R$ **do**
4:   $f_r \leftarrow \mathrm{FMS}_{\mathrm{avg}}(\beta_r)$.  // `Eq.(9)`
5: **end for**
6: $\tau_r \leftarrow (1 - \beta_r)^{-1}$.
7: $\phi_r \leftarrow \mathrm{Norm}(\log \tau_r); \quad \psi_r \leftarrow \mathrm{Norm}(f_r)$.
8: $\widehat{L}(\phi_r) \leftarrow (1 - \phi_r)\,\psi_1 + \phi_r\,\psi_R$.
9: **for** $r = 1$ to $R$ **do**
10:   $d_r \leftarrow \left| \widehat{L}(\phi_r) - \psi_r \right|$.
11: **end for**
12: $r^* \leftarrow \arg\max_r d_r$.
13: $\beta^\dagger \leftarrow \beta_{r^*}$.
14: **Output:** reference boundary $\beta^\dagger$.

---

dataset's discriminative spectrum to quantify mechanism-data alignment.

### 3.4. Frequency-Guided Membrane Decay Criterion

High-frequency bands often entangle high-$\mathrm{DI}(\omega_k)$ cues with interference, requiring a trade-off between information retention and noise suppression. Motivated by this finding, we leverage the $\beta$-controlled low-pass bias of LIF dynamics to selectively attenuate high-frequency noise without hard truncation. We next formalize this data–dynamics matching and derive actionable criteria for choosing $\beta$.

On the one-sided frequency grid $\Omega_L$, we quantify this alignment using the Frequency-Matching Score (FMS):

$$\mathrm{FMS}_{\mathrm{avg}}(\beta) = \sum_{\omega_k \in \Omega_L} \mathrm{DI}_{\mathrm{norm}}(\omega_k)\, \tilde{H}(\omega_k; \beta) \qquad (9)$$

The core idea of $\mathrm{FMS}_{\mathrm{avg}}(\beta)$ is to measure mechanism–data alignment in the frequency domain by measuring how well the LIF neuron preserves the dataset's discriminative spectral content. Concretely, on the shared one-sided grid $\omega_k \in \Omega_L$, $\mathrm{DI}_{\mathrm{norm}}(\omega_k)$ assigns a *class-separability* weight to each frequency bin (i.e., the normalized discriminative spectrum), computed once from the training fold and then held fixed for all $\beta$. In contrast, $\tilde{H}(\omega_k; \beta)$ is the LIF DC-normalized frequency-response magnitude at $\omega_k$, whose low-pass strength is controlled by $\beta$. We form per-bin contributions via the product $\mathrm{DI}_{\mathrm{norm}}(\omega_k)\tilde{H}(\omega_k; \beta)$ and sum them over $\Omega_L$ to obtain $\mathrm{FMS}_{\mathrm{avg}}(\beta)$ (see Figure 2(c)). Since $\tilde{H}(\omega_k; \beta) \in [0, 1]$, it follows that $\mathrm{FMS}_{\mathrm{avg}}(\beta) \in [0, 1]$, which admits a simple interpretation: for a given $\beta$, it is the class-weighted fraction of discriminative spectral content retained by the LIF low-pass mechanism.

Therefore, in our analysis, $\mathrm{FMS}_{\mathrm{avg}}(\beta)$ is chiefly controlled by the leak parameter $\beta$. Although one might posit an opti-

mal $\beta$ that maximizes mechanism–data alignment, accuracy-based tuning of $\beta$ typically relies on costly dataset-specific sweeps and offers limited mechanistic insight. Instead of tuning $\beta$ against accuracy, we define a fully specified, lower-bound–like *reference boundary* on $\beta$ that marks the onset of over-low-pass behavior.

**Definition 3.4** (Maximum-deviation rule). Given a discrete candidate set $\mathcal{B} = \{\beta_r\}_{r=1}^R$, we re-parameterize each $\beta_r$ by $\tau_r \triangleq (1 - \beta_r)^{-1}$ and construct normalized coordinates

$$\phi_r \triangleq \mathrm{Norm}\big(\log \tau_r\big) \in [0, 1], \; r \in [1, R] \qquad (10)$$

$$\psi_r \triangleq \mathrm{Norm}\big(\mathrm{FMS}_{\mathrm{avg}}(\beta_r)\big) \in [0, 1], \; r \in [1, R] \qquad (11)$$

where $\mathrm{Norm}(\cdot)$ is a fixed linear scaling over the sampled points (e.g., min-max), and all edge cases (degenerate ranges) and deterministic selection rules are fixed in advance. We use $\log \tau$ because $\tau$ spans orders of magnitude as $\beta \to 1$; measuring deviation in $\log \tau$ therefore linearizes the sweep and prevents the high-$\beta$ regime from dominating the selection of $\beta^\dagger$. Let $\widehat{L}(\cdot)$ denote the reference diagonal defined by the line segment connecting the two endpoints in the normalized coordinate system, then we have:

$$\widehat{L}(\phi_r) \triangleq (1 - \phi_r)\,\psi_1 + \phi_r\,\psi_R, r \in [1, R] \qquad (12)$$

We define the deviation score $d(\beta_r)$ as the vertical distance from the reference diagonal $\widehat{L}$ (Satopaa et al., 2011) as:

$$d(\beta_r) \triangleq \left| \widehat{L}(\phi_r) - \psi_r \right|, \; \beta^\dagger \triangleq \arg\max_{\beta_r \in \mathcal{B}} d(\beta_r) \qquad (13)$$

As depicted in Figure 2(d), the maximum-deviation rule provides a fully specified, accuracy-agnostic method to identify a *reference boundary* on the $\beta$ axis, derived solely from the geometry of the $\mathrm{FMS}_{\mathrm{avg}}(\beta)$ curve under increasing low-pass strength. Importantly, $\beta^\dagger$ is not intended to optimize performance; it indicates the point at which further low-pass bias produces diminishing retention of the discriminative spectrum. This deterministic boundary enables a mechanism-centric partitioning of $\beta$ into qualitatively distinct operating regimes, as summarized next:

**Proposition 3.5** (Three $\beta$ regimes (mechanism at a glance)). *Let* $\mathrm{FMS}_{\mathrm{avg}}(\beta)$ *denote the dataset-level* discriminative-spectrum retention *induced by the DC-normalized LIF low-pass template* $\tilde{H}(\cdot; \beta)$. *Let* $\beta^\dagger$ *be a deterministic* reference (risk) boundary *that marks the onset of* over-low-pass behavior. *Then* $\beta$ *admits three mechanism-centric regimes:*

(i) ***Under-filter*** *($\beta \to 0$): $\tilde{H}(\cdot; \beta)$ is near-passband, so $\mathrm{FMS}_{\mathrm{avg}}(\beta)$ is high (approaching 1); nuisance suppression is weak, and performance can be less stable.*

(ii) ***Stability window*** *($0 < \beta < \beta^\dagger$): $\mathrm{FMS}_{\mathrm{avg}}(\beta)$ decreases moderately as the low-pass bias becomes effective; accuracy may peak in this region due to nuisance suppression/optimization stability.*

*Table 1.* Overall test accuracy (%) and parameter count (M) of baselines across datasets (mean over three random seeds). **Bold** denotes the best and underline denotes the second-best on each dataset.

| Baseline | AOPHand | | mmFiT | | Pantomime | | MMActivity | |
|---|---|---|---|---|---|---|---|---|
| | Accuracy (%) | #Params (M) | Accuracy (%) | #Params (M) | Accuracy (%) | #Params (M) | Accuracy (%) | #Params (M) |
| MLP | $69.70_{\pm 0.99}$ | 4.327 | $66.51_{\pm 2.02}$ | 4.327 | $66.81_{\pm 0.57}$ | 4.328 | $60.83_{\pm 2.89}$ | 4.327 |
| LeNet | $60.86_{\pm 1.68}$ | 4.191 | $62.36_{\pm 0.58}$ | 4.192 | $61.83_{\pm 0.42}$ | 4.196 | $59.17_{\pm 3.82}$ | 4.191 |
| VGG9 | $\underline{74.39}_{\pm 0.61}$ | 31.603 | $69.36_{\pm 2.15}$ | 31.605 | $72.63_{\pm 0.46}$ | 31.622 | $70.00_{\pm 4.33}$ | 31.601 |
| VGG16 | $67.92_{\pm 6.02}$ | 70.313 | $65.43_{\pm 0.58}$ | 70.315 | $71.60_{\pm 1.88}$ | 70.331 | $62.50_{\pm 4.33}$ | 70.311 |
| ResNet18 | $72.47_{\pm 1.30}$ | 11.173 | $71.94_{\pm 1.29}$ | 11.174 | $72.63_{\pm 0.59}$ | 11.178 | $61.67_{\pm 6.29}$ | 11.173 |
| ResNet50 | $72.54_{\pm 0.80}$ | 23.514 | $71.84_{\pm 1.97}$ | 23.516 | $73.90_{\pm 0.43}$ | 23.532 | $61.67_{\pm 2.89}$ | 23.512 |
| ResNet101 | $69.70_{\pm 0.72}$ | 42.506 | $\underline{72.64}_{\pm 0.25}$ | 42.508 | $73.85_{\pm 0.29}$ | 42.525 | $64.17_{\pm 1.44}$ | 42.504 |
| RNN | $26.40_{\pm 9.25}$ | **0.026** | $22.56_{\pm 5.93}$ | **0.026** | $20.84_{\pm 0.88}$ | **0.027** | $48.33_{\pm 1.44}$ | **0.025** |
| GRU | $67.52_{\pm 5.73}$ | 0.075 | $14.11_{\pm 0.38}$ | 0.075 | $\underline{75.45}_{\pm 0.99}$ | 0.076 | $47.50_{\pm 2.50}$ | 0.075 |
| LSTM | $20.71_{\pm 1.12}$ | 0.100 | $13.14_{\pm 0.52}$ | 0.100 | $72.47_{\pm 3.29}$ | 0.101 | $54.17_{\pm 7.64}$ | 0.100 |
| BiLSTM | $46.14_{\pm 1.40}$ | 0.200 | $12.98_{\pm 0.89}$ | 0.200 | $73.67_{\pm 1.39}$ | 0.203 | $45.83_{\pm 3.82}$ | 0.200 |
| CNN-GRU | $61.98_{\pm 9.44}$ | 0.463 | $67.80_{\pm 0.49}$ | 0.463 | $72.77_{\pm 2.36}$ | 0.464 | $65.00_{\pm 2.50}$ | 0.463 |
| ViT | $21.39_{\pm 3.31}$ | 2.176 | $36.40_{\pm 5.65}$ | 2.176 | $42.16_{\pm 6.07}$ | 2.178 | $65.83_{\pm 5.77}$ | 2.175 |
| **SpikingLeNet** | $\mathbf{83.70}_{\pm 4.24}$ | 4.191 | $\mathbf{73.67}_{\pm 1.55}$ | 4.192 | $\mathbf{78.31}_{\pm 0.50}$ | 4.196 | $\mathbf{75.00}_{\pm 6.61}$ | 4.191 |

*(iii)* ***Over-low-pass*** $(\beta \geq \beta^{\dagger})$*:* $\mathrm{FMS}_{\mathrm{avg}}(\beta)$ *becomes low, indicating loss of non-DC discriminative content; accuracy degradations are empirically more frequent.*

As summarized in Algorithm 1, we first characterize (i) where discriminative information resides in mmWave data via $\mathrm{DI}_{\mathrm{norm}}$ and (ii) how LIF neurons selectively preserve frequency components via $\tilde{H}(\cdot; \beta)$, with the half-power cutoff defining an effective bandwidth $B_{\mathrm{eff}}(\beta)$ that acts as a principled bandwidth control. Their interaction defines the frequency-matching score $\mathrm{FMS}_{\mathrm{avg}}(\beta)$, quantifying the alignment between mmWave discriminative content and LIF's low-pass bias. For high-frequency ranges containing both noise and informative signal, $\beta$ mediates a trade-off between noise suppression and signal retention. To capture mechanism–data alignment independently of accuracy, we define a reference boundary $\beta^{\dagger}$ that marks the onset of over-low-pass behavior and serves as an anchor for regime analysis and evaluation.

## 4. Experiments

### 4.1. Experimental Settings

**Datasets.** We conduct experiments on *four* widely used mmWave-based wireless sensing datasets: *AOPHand* (Zafar, 2023), *mmFiT* (Tiwari, 2023), *Pantomime* (Palipana et al., 2021), and *MMActivity* (Singh et al., 2019). For all datasets, we use a uniform 80/20 train–test split. Further details on the datasets and their preprocessing protocols are provided in ***Appendix D***.

**Baselines.** We use a simple LeNet-style SNN, termed *SpikingLeNet*, as our primary baseline and compare it against a broad set of ANN models commonly used in wireless sensing (Yang et al., 2023), including *MLP*, convolutional models (*LeNet*, *VGG*), residual networks (*ResNet*), recur-

*Table 2.* Theoretical energy cost ($\mu$J) of baselines across datasets.

| Baseline | AOPHand | mmFiT | Pantomime | MMActivity |
|---|---|---|---|---|
| MLP | 19.90 | 19.90 | 19.91 | 19.90 |
| LeNet | 251.08 | 251.08 | 251.10 | 251.08 |
| VGG9 | 3352.70 | 3352.71 | 3352.79 | 3352.69 |
| VGG16 | 6017.25 | 6017.26 | 6017.34 | 6017.24 |
| ResNet18 | 655.24 | 655.22 | 655.24 | 655.22 |
| ResNet50 | 1522.02 | 1522.02 | 1522.10 | 1522.01 |
| ResNet101 | 2923.68 | 2923.69 | 2923.76 | 2923.67 |
| RNN | 7.35 | 7.35 | 7.36 | 7.35 |
| GRU | 22.20 | 22.20 | 22.20 | 22.20 |
| LSTM | 29.55 | 29.55 | 29.55 | 29.54 |
| BiLSTM | 59.09 | 59.10 | 59.10 | 59.10 |
| CNN-GRU | 128.55 | 128.55 | 128.56 | 128.55 |
| ViT | 82.61 | 82.61 | 82.62 | 82.61 |
| **SpikingLeNet** | **2.53** | **2.04** | **2.44** | **1.45** |

rent models (*RNN*, *GRU*, *LSTM*, *BiLSTM*), *CNN-GRU*, and vision transformers (*ViT*). To ensure fair comparison, all models are evaluated under a unified training and evaluation protocol (***Appendix E***).

**Metrics.** We assess the advantages of SNNs over ANNs on mmWave edge sensing tasks using multiple metrics, including top-1 accuracy, model parameter count, end-to-end inference latency, number of operations per sample, and theoretical energy consumption per sample. Details of the measurement protocol are provided in ***Appendix F***.

### 4.2. Overall Performance

Table 1 summarizes the overall recognition accuracy of SpikingLeNet compared with ANN baselines. Across all datasets, SpikingLeNet consistently outperforms the strongest ANN counterparts, achieving an average accuracy improvement of approximately 6.22%. Notably, these gains are achieved with the same parameter budget as LeNet

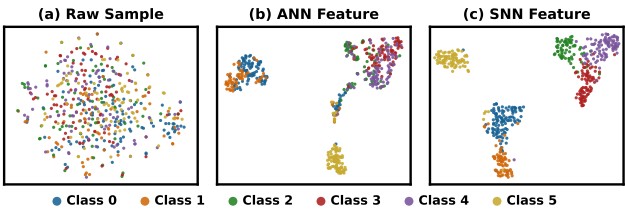

*Figure 3.* t-SNE visualization of raw inputs from AOPHand and the corresponding features extracted from the penultimate layer by ANN and SNN with the same LeNet backbone.

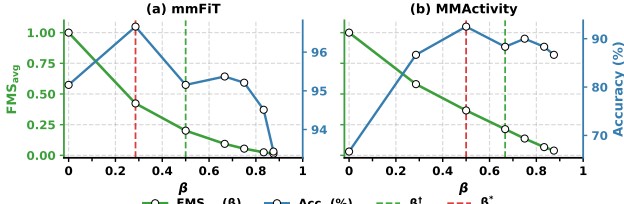

*Figure 4.* Accuracy–FMS alignment under $\beta$ tuning. Increasing $\beta_r$ leads to a monotonic decrease in $\text{FMS}_{\text{avg}}$, while accuracy is non-monotonic and attains its maximum at $\beta^* < \beta^\dagger$.

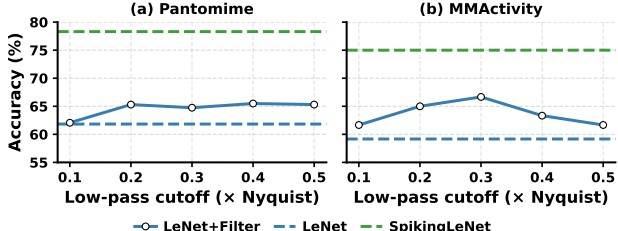

*Figure 5.* Impact of low-pass filter preprocessing with varying cutoff frequencies (normalized by Nyquist) for LeNet-based ANNs versus SpikingLeNet. See *Appendix I* for implementation details.

($\approx$4.19M) and substantially fewer parameters than high-capacity CNNs such as VGG9 (31.6M) and ResNet101 (42.5M), indicating superior parameter efficiency. Moreover, SpikingLeNet exhibits more stable performance across datasets, whereas several ANN baselines suffer pronounced degradation on specific datasets (e.g., RNN/GRU on mmFiT). Since these improvements cannot be attributed solely to model capacity, they are consistent with our hypothesis that LIF dynamics introduce an implicit low-pass temporal integration that better matches the spectral characteristics of mmWave signals. Performance comparisons with *a broader set of SNN architectures* are reported in *Appendix G*, further corroborating the proposed frequency-matching hypothesis.

### 4.3. Efficiency Analysis

As reported in Table 2, SpikingLeNet is consistently the most energy-efficient model across all datasets. Among ANN baselines, the best-performing model is the RNN, which consumes 7.35–7.36 $\mu$J per sample, whereas SpikingLeNet requires only 1.45–2.53 $\mu$J. This corresponds to an average reduction of $\sim$3.64$\times$ (approximately $\sim$75% lower) when compared with the per-dataset best ANN. The advantage becomes substantially more pronounced relative to mainstream CNN and Transformer baselines: LeNet consumes about 251 $\mu$J ($\sim$119$\times$ higher than SpikingLeNet on average), while ResNet and VGG variants span $\approx$655 to $\approx$6017 $\mu$J, corresponding to energy costs that are hundreds to thousands of times higher. Owing to the event-driven nature of SNN inference, sparse spike activations replace dense MAC-heavy floating-point operations, enabling temporal integration with far fewer high-cost computations than conventional ANN forward passes. Detailed computing operation statistics for theoretical energy cost analysis are provided in *Appendix H*.

### 4.4. Mechanistic Analysis of Frequency Matching

**t-SNE Visualization.** Figure 3 presents a t-SNE (Maaten & Hinton, 2008) visualization of the learned embeddings after the local update for the 6-class AOPHand task, where tighter intra-class clusters and larger inter-class margins indicate higher discriminability. ANN features exhibit pronounced overlap, particularly among Classes 2–4, suggesting am-

biguous decision boundaries. In contrast, SNN embeddings form substantially more compact clusters with clearer class separation. We attribute this improvement to a closer alignment between the spectral–temporal structure of mmWave signals and the frequency-selective dynamics of spiking neurons, which suppresses nuisance high-frequency noise while preserving discriminative components. This mechanism yields more separable feature representations than those produced by dense-computation ANNs.

**Accuracy-FMS Alignment.** To verify our proposed membrane decay criteria, we examine the correlation between accuracy and $\text{FMS}_{\text{avg}}$ across different $\beta$ settings in Figure 4. The observations illustrate that as $\beta$ increases, $\text{FMS}_{\text{avg}}$ decreases approximately monotonically from near 1 toward 0, indicating progressively stronger low-pass attenuation induced by LIF dynamics. In contrast, accuracy is distinctly non-monotonic: it improves at moderate $\beta$ and then deteriorates under overly strong temporal smoothing. On Figure 4(a), accuracy increases to a clear maximum at an intermediate $\beta^*$ and then declines for large $\beta$; on Figure 4(b), a similar rise-and-fall trend is observed. Importantly, the optimal setting $\beta^*$ consistently occurs *before* the reference boundary $\beta^\dagger$ on both datasets, in agreement with Proposition 3.5. This behavior admits a spectral explanation: increasing $\beta$ strengthens the effective LIF low-pass filtering and suppresses high-frequency components, which is beneficial initially due to substantial high-frequency noise in mmWave signals; however, discriminative cues are not confined to the lowest band, so excessive attenuation removes class-relevant details and degrades recognition.

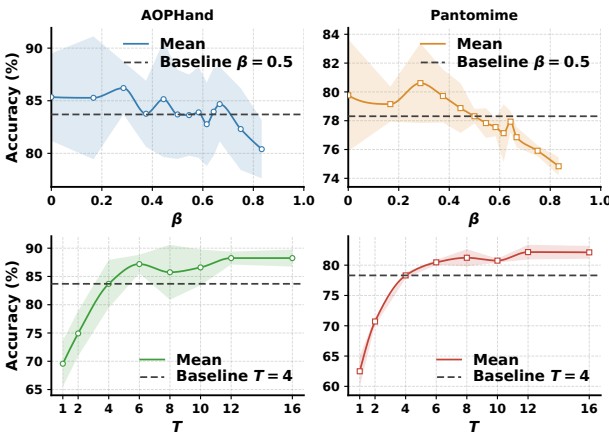

*Figure 6.* Ablation of SNN hyperparameters.

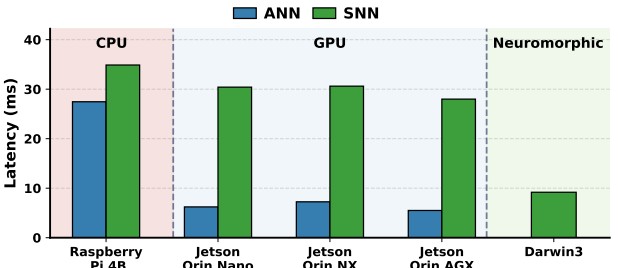

*Figure 7.* Per-sample on-device inference latency (ms).

### 4.5. Ablation Study

**Impact of Low-pass Filter.** To isolate the effect of LIF's intrinsic low-pass dynamics, we compare SpikingLeNet with a LeNet augmented by an explicit low-pass filter. As shown in Figure 5, the filter improves LeNet relative to its unfiltered version but still underperforms SpikingLeNet, since it indiscriminately removes all spectral components above the cutoff, discarding high-frequency discriminative cues along with noise. In contrast, LIF dynamics provide a tunable low-pass filter that selectively attenuates high-frequency content, achieving a superior trade-off between noise suppression and information preservation, and consequently higher accuracy.

**Hyper-Parameters.** Figure 6 ablates two key SNN hyperparameters, the membrane decay factor $\beta$ and the number of simulation timesteps $T$, on AOPHand and Pantomime. Test accuracy shows a clear peak as a function of $\beta$: performance improves up to a dataset-specific optimum and then declines. This aligns with our frequency-matching hypothesis: optimal accuracy occurs when LIF-induced low-pass dynamics suppress high-frequency noise while preserving task-relevant spectral components. By contrast, increasing $T$ boosts accuracy at small values but quickly saturates, indicating diminishing returns from longer temporal integration. Overall, these results confirm that SNN performance is driven by temporal dynamics: $\beta$ should match the dataset's spectral structure, while $T$ mainly stabilizes predictions.

### 4.6. Practical Implementation

Figure 7 reports inference latency measured on five deployment platforms. On GPU-equipped Jetson devices, SpikingLeNet incurs an approximately $4\times$ latency overhead relative to LeNet, consistent with executing a similar feed-forward graph over 4 simulation timesteps ($T{=}4$) when spiking operations are realized as dense GPU kernels. This scaling does not fully transfer to the CPU-only Rasp-

berry Pi, where SpikingLeNet is only marginally slower than LeNet, as latency is dominated by system-level factors such as control flow and memory bandwidth rather than raw MAC throughput. In contrast, on the neuromorphic chip Darwin3 (Ma et al., 2024), SpikingLeNet achieves latency comparable to LeNet on conventional edge hardware, highlighting the benefit of hardware–algorithm co-design: event-driven execution can exploit spike sparsity and avoid redundant computation. Overall, these results indicate that the current latency gap between SNNs and mainstream accelerators is largely a systems artifact, and that neuromorphic platforms offer a promising path toward efficient, real-time mmWave inference. ***Appendix J*** provides device specifications and the full latency measurement protocol.

## 5. Related Work

### 5.1. Spiking Neural Networks for mmWave Sensing

Existing SNN-based mmWave studies largely fall into two categories. The first (Wu et al., 2025; Li et al., 2024; Zhang et al., 2024; Arsalan et al., 2022a) treats SNNs as energy-efficient alternatives to ANNs, demonstrating theoretical low-power and low-latency benefits on edge mmWave tasks such as gesture and gait recognition. The second (Hu et al., 2025; Shaaban et al., 2024; Arsalan et al., 2023; Liu et al., 2022) focuses on improving recognition accuracy, typically through spike encoding strategies, training recipes, and extensive hyperparameter tuning. However, these prior studies primarily emphasize metric improvements and offer limited insight into how spiking dynamics interact with mmWave signals, leaving it unclear under what conditions SNNs surpass ANNs. Addressing this gap is critical for designing SNNs that achieve both high accuracy and energy efficiency in mmWave sensing.

### 5.2. Mechanistic Analysis of Spiking Neural Networks

Prior work has also pursued a mechanistic understanding of SNNs. One line of research (Fang et al., 2025; Mani et al., 2025; Kiessling & Lindner, 2024; Kim et al., 2023) interprets individual spiking neurons or shallow SNNs through the lens of explainable algorithmic and signal-processing operators, clarifying how membrane integration and thresh-

olded spiking impose specific inductive biases and shape information flow. Another line of analysis (Lv et al., 2025; Zhang et al., 2025; Cao et al., 2024; Lv et al., 2024a) examines how SNNs represent sequential structure and positional information, thereby motivating mechanism-aware designs for sequence modeling. However, most mechanistic studies remain task-centric: they characterize the biases induced by spiking dynamics but rarely connect them to modality-specific signal structure. As a result, it remains difficult to anticipate when and why SNNs outperform ANNs in particular sensing modalities, motivating our interpretable mechanism–data alignment framework.

## 6. Conclusion

We present a mechanistic frequency-domain analysis of SNNs for mmWave sensing, linking mmWave spectral structure with the intrinsic low-pass dynamics of LIF neurons to explain when and why spiking models are effective. This alignment yields a deterministic, accuracy-agnostic guideline for configuring the membrane decay factor to match neuronal bandwidth to discriminative frequency content. Extensive experiments validate the frequency-matching hypothesis, demonstrating consistent accuracy and energy-efficiency gains over ANN baselines. Limitations and future work are discussed in *Appendix K*.

## Impact Statement

This work advances the understanding and deployment of spiking neural networks for mmWave sensing. By characterizing the interaction between dataset-specific spectral structure and LIF temporal dynamics, we provide actionable guidance for configuring SNNs to improve accuracy and energy efficiency on edge hardware, thereby reducing the compute and power requirements of real-time sensing systems. Besides, we do not think our work will have a negative impact on ethical considerations or future societal consequences.

## Acknowledgements

The work of this paper is supported by the National Key Research and Development Program of China under Grant No. 2022YFB4500100, the National Natural Science Foundation of China under Grant No. 62502443, No. 62125206, and the Zhejiang Provincial Natural Science Foundation of China under Grant No. LD24F020014.

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

# A. DI Estimation Details and Robustness

This appendix specifies the *train-only* pipeline used to estimate the dataset-level discriminative spectrum $\text{DI}_{\text{norm}}(\omega_k)$. Throughout, we strictly follow the notation in the symbol table and *only compare quantities defined on the same one-sided grid*

$$\Omega_L = \left\{ \omega_k = \frac{2\pi k}{L} \;\middle|\; k = 0, 1, \dots, K \right\}, \qquad K = \left\lfloor \frac{L}{2} \right\rfloor. \tag{14}$$

This one-sided construction retains the DC bin ($k = 0$) and all nonnegative DFT frequencies up to the largest one-sided index $K = \lfloor L/2 \rfloor$. When $L$ is even, $K = L/2$ corresponds to the Nyquist bin at $\omega = \pi$, which is included exactly once in $\Omega_L$. When $L$ is odd, there is no Nyquist bin; the highest retained frequency is $k = (L-1)/2$, i.e., $\omega_K = 2\pi K/L < \pi$.

## A.1. Deterministic Input Processing and Scalarization Scope

### A.1.1. DATA AND INDEXING

We start from the supervised mmWave dataset

$$\mathcal{D} = \{(\mathbf{X}_i, y_i)\}_{i=1}^{M}, \qquad y_i \in \{1, \dots, \mathcal{C}\}. \tag{15}$$

Each sample is a real-valued tensor with an explicit temporal axis of length $L$:

$$\mathbf{X}_i \in \mathbb{R}^{L \times C \times H \times W}, \qquad \mathbf{X}_i[l] \in \mathbb{R}^{C \times H \times W}, \;\; l \in \{0, \dots, L-1\}. \tag{16}$$

All DI statistics are computed *once* using only the training split $\mathcal{D}_{\text{tr}} \subseteq \mathcal{D}$ and then frozen (cf. §A.3). For implementation alignment, we use $0$-based indexing for $l$ and for the one-sided bin index $k$; all definitions are invariant to this convention.

### A.1.2. SCALARIZATION AND SCOPE

To make this comparison well-defined across heterogeneous mmWave layouts, we reduce each frame to a scalar by averaging over *all* non-temporal axes:

$$\mathbf{s}_i[l] \triangleq \frac{1}{CHW} \sum_{c=1}^{C} \sum_{h=1}^{H} \sum_{w=1}^{W} \mathbf{X}_i[l, c, h, w] \in \mathbb{R}. \tag{17}$$

This aggregation is intentionally non-invertible (it discards spatial/channel correlations).

### A.1.3. PREPROCESSING

Given the scalar temporal sequence $\{\mathbf{s}_i[l]\}_{l=0}^{L-1}$, we form a preprocessed sequence $\{\bar{\mathbf{s}}_i[l]\}_{l=0}^{L-1}$ by a deterministic rule:

$$\bar{\mathbf{s}}_i[l] = \begin{cases} \mathbf{s}_i[l], & \texttt{raw}, \\ \mathbf{s}_i[l] - \frac{1}{L} \sum_{l'=0}^{L-1} \mathbf{s}_i[l'], & \texttt{demean}. \end{cases} \tag{18}$$

Here $l'$ denotes a dummy index used to compute the temporal mean over the sequence. The `raw` mode uses the scalar proxy as-is. The `demean` mode subtracts the temporal mean, thereby removing the DC component (zero-frequency offset) before taking the DFT.

## A.2. One-Sided DFT on the Common Grid with Optional Windowing

### A.2.1. GRID AND ONE-SIDED TRANSFORM

We compute spectra on the fixed one-sided grid $\Omega_L$ in Eq. (14). For each real-valued sequence $\bar{\mathbf{s}}_i[l] \in \mathbb{R}$ of length $L$, we apply the temporal DFT operator $\mathcal{F}_t(\cdot)$ and denote its coefficients by

$$\tilde{\mathbf{s}}_i[k] \triangleq \big(\mathcal{F}_t(\bar{\mathbf{s}}_i)\big)(\omega_k), \qquad \omega_k = \frac{2\pi k}{L}, \quad k = 0, 1, \dots, K. \tag{19}$$

We then use the nonnegative amplitude feature (discarding phase):

$$A_i[k] \triangleq |\tilde{\mathbf{s}}_i[k]| \in \mathbb{R}_{\geq 0}. \tag{20}$$

Because $\bar{s}_i[l]$ is real, the full DFT is conjugate-symmetric; thus the bins $k = 0, \ldots, K$ correspond to the nonnegative-frequency half. DC ($k = 0$) appears once, and the Nyquist bin ($k = K$ when $L$ is even) also appears once, consistent with the fixed one-sided convention.

**Normalization and scale.**  Our implementation uses an unnormalized DFT. Any constant scaling factor applied to $\tilde{s}_i[k]$ induces the same global scaling on all $A_i[k]$, which cancels in the DI ratio up to the stabilizer $\varepsilon$ (see §A.3 and §A.7).

### A.2.2. OPTIONAL WINDOWING

By default, estimation uses the implicit rectangular window. To reduce spectral leakage, we also test a Hann-windowed variant applied *after* preprocessing and *before* the DFT:

$$\bar{s}_i^{(w)}[l] \triangleq w[l]\bar{s}_i[l], \tag{21}$$

$$w[l] = \tfrac{1}{2}\left(1 - \cos\frac{2\pi l}{L-1}\right), \qquad l = 0, \ldots, L-1, \tag{22}$$

and then replace $\bar{s}_i[l]$ with $\bar{s}_i^{(w)}[l]$ in Eq. (19). All downstream definitions remain unchanged and still live on the same $\Omega_L$ grid.

## A.3. DI Estimator and Normalization

### A.3.1. TRAIN-ONLY CLASS STATISTICS

All statistics are computed *once* on the stratified training split $\mathcal{D}_{\mathrm{tr}}$ and then frozen. Let $\mathcal{I}_{\mathrm{tr}}$ index the training samples and define

$$N_{\mathrm{tr}} \triangleq |\mathcal{I}_{\mathrm{tr}}|, \tag{23}$$

$$N_{c,\mathrm{tr}} \triangleq \left|\{i \in \mathcal{I}_{\mathrm{tr}} : y_i = c\}\right|, \tag{24}$$

$$\pi_c \triangleq \frac{N_{c,\mathrm{tr}}}{N_{\mathrm{tr}}}. \tag{25}$$

For each one-sided bin $k \in \{0, \ldots, K\}$, define the class-conditional mean of the amplitude feature and the mixture mean:

$$\mu_c[k] \triangleq \frac{1}{N_{c,\mathrm{tr}}} \sum_{\substack{i \in \mathcal{I}_{\mathrm{tr}} \\ y_i = c}} A_i[k], \tag{26}$$

$$\bar{\mu}[k] \triangleq \sum_{c=1}^{\mathcal{C}} \pi_c\, \mu_c[k]. \tag{27}$$

Next, define the unbiased within-class variance (well-defined in our stratified splits for all reported datasets, where $N_{c,\mathrm{tr}} \geq 2$ holds):

$$\mathrm{Var}_c[k] \triangleq \frac{1}{N_{c,\mathrm{tr}} - 1} \sum_{\substack{i \in \mathcal{I}_{\mathrm{tr}} \\ y_i = c}} \left(A_i[k] - \mu_c[k]\right)^2. \tag{28}$$

### A.3.2. BETWEEN/WITHIN SCATTERS AND DI

Following the symbol table definitions, the between-class and within-class scatters at bin $k$ are

$$S_{\mathrm{B}}[k] \triangleq \sum_{c=1}^{\mathcal{C}} \pi_c\left(\mu_c[k] - \bar{\mu}[k]\right)^2, \tag{29}$$

$$S_{\mathrm{W}}[k] \triangleq \sum_{c=1}^{\mathcal{C}} \pi_c\, \mathrm{Var}_c[k]. \tag{30}$$

We then define the discriminative index at $\omega_k$ as the stabilized ratio (Fukunaga, 2013)

$$\mathrm{DI}(\omega_k) \triangleq \frac{\mathrm{S_B}[k]}{\mathrm{S_W}[k] + \varepsilon}, \qquad \varepsilon > 0. \tag{31}$$

The stabilizer $\varepsilon$ prevents division by near-zero $\mathrm{S_W}[k]$ (e.g., extremely small within-class variability at some frequencies).

### A.3.3. NORMALIZATION ON $\Omega_L$

To compare against in discrete-time frequency-matching Analysis (DFMA), we normalize DI over the fixed one-sided grid:

$$\mathrm{DI}_{\mathrm{norm}}(\omega_k) \triangleq \frac{\mathrm{DI}(\omega_k)}{\sum_{k'=0}^{K} \mathrm{DI}(\omega_{k'})}. \tag{32}$$

By construction, $\mathrm{DI}_{\mathrm{norm}}(\omega_k) \geq 0$ and $\sum_{k=0}^{K} \mathrm{DI}_{\mathrm{norm}}(\omega_k) = 1$, so it forms a probability mass function (PMF) (Kokonendji et al., 2007) over $\Omega_L$. This is the exact object used in the DFMA score $\mathrm{FMS}_{\mathrm{avg}}(\beta) = \sum_{k=0}^{K} \mathrm{DI}_{\mathrm{norm}}(\omega_k) \tilde{H}(\omega_k; \beta)$.

**No leakage.** To prevent leakage, $\mathrm{DI}(\omega_k)$ and $\mathrm{DI}_{\mathrm{norm}}(\omega_k)$ are computed *only* from $\mathcal{D}_{\mathrm{tr}}$ and then frozen for all downstream analyses and for all candidate $\beta \in \mathcal{B}$. In particular, we never compute DI using the test split and never use test-derived DI statistics for any model or hyperparameter selection.

### A.4. Fisher-Style Derivation of the Per-Frequency DI

**Setup: treat each frequency bin as a scalar feature.** Fix a frequency bin $k$ (equivalently $\omega_k$). Define the scalar random variable

$$X_k \in \mathbb{R}, \tag{33}$$

whose empirical realizations are $\{A_i[k]\}_{i \in \mathcal{I}_{\mathrm{tr}}}$ on the training split, with class label $y_i \in \{1, \ldots, \mathcal{C}\}$. Throughout this derivation, *all expectations are class-prior weighted* by $\pi_c$ (defined in Eq. (25)).

**Class means and variances.** For each class $c$, define the (population) class-conditional mean and variance of $X_k$ as

$$m_c[k] := \mathbb{E}[X_k \mid y = c], \tag{34}$$
$$\sigma_c^2[k] := \mathrm{Var}(X_k \mid y = c), \tag{35}$$

and define the mixture mean

$$m[k] := \sum_{c=1}^{\mathcal{C}} \pi_c\, m_c[k] = \mathbb{E}[X_k]. \tag{36}$$

In our implementation, we estimate these from the training data via

$$m_c[k] \rightsquigarrow \mu_c[k], \tag{37}$$
$$m[k] \rightsquigarrow \bar{\mu}[k], \tag{38}$$
$$\sigma_c^2[k] \rightsquigarrow \mathrm{Var}_c[k], \tag{39}$$

where $\rightsquigarrow$ denotes replacement by the corresponding empirical estimator computed on the training split, and $\mu_c[k], \bar{\mu}[k], \mathrm{Var}_c[k]$ are exactly those in Eq. (26)–Eq. (28).

**Fisher criterion (general form) and its 1D specialization.** The classical Fisher criterion measures class separability (Murphy, 2022) after a linear projection. For a $d$-dimensional feature vector $z \in \mathbb{R}^d$ and projection $w \in \mathbb{R}^d$, one commonly writes the (scalar) objective as the Rayleigh quotient

$$\mathcal{J}(w) := \frac{w^\top S_B w}{w^\top S_W w}, \tag{40}$$

where $S_B$ and $S_W$ are between-class and within-class scatter matrices.

In our case, for each fixed bin $k$, the feature is already *one-dimensional*:

$$z := X_k \in \mathbb{R} \quad (d = 1), \qquad w \in \mathbb{R}. \tag{41}$$

Thus, Eq. (40) *specializes to a scalar ratio* because

$$w^\top S_B w = w^2 \, S_B^{(1D)}[k], \tag{42}$$

$$w^\top S_W w = w^2 \, S_W^{(1D)}[k], \tag{43}$$

and hence for any $w \neq 0$,

$$\mathcal{J}(w) = \frac{w^2 S_B^{(1D)}[k]}{w^2 S_W^{(1D)}[k]} = \frac{S_B^{(1D)}[k]}{S_W^{(1D)}[k]}. \tag{44}$$

Therefore, in 1D the Fisher criterion is *projection-invariant* (up to the sign/scale of $w$) and reduces to the *between/within scatter ratio* of the scalar feature itself.

**Between-class scatter in 1D (population form).**  Define the 1D between-class scatter for bin $k$ as the prior-weighted variance of class means:

$$S_B^{(1D)}[k] := \sum_{c=1}^{\mathcal{C}} \pi_c \big( m_c[k] - m[k] \big)^2. \tag{45}$$

Expanding the square yields an equivalent closed form:

$$
\begin{aligned}
S_B^{(1D)}[k] &= \sum_{c=1}^{\mathcal{C}} \pi_c \big( m_c[k]^2 - 2 m_c[k] m[k] + m[k]^2 \big) \\
&= \sum_{c=1}^{\mathcal{C}} \pi_c m_c[k]^2 - 2 m[k] \sum_{c=1}^{\mathcal{C}} \pi_c m_c[k] + m[k]^2 \sum_{c=1}^{\mathcal{C}} \pi_c \\
&= \sum_{c=1}^{\mathcal{C}} \pi_c m_c[k]^2 - 2 m[k] \cdot m[k] + m[k]^2 \\
&= \sum_{c=1}^{\mathcal{C}} \pi_c m_c[k]^2 - m[k]^2.
\end{aligned} \tag{46}
$$

The second line uses Eq. (36) and $\sum_c \pi_c = 1$.

**Within-class scatter in 1D (population form).**  Define the 1D within-class scatter for bin $k$ as the prior-weighted average within-class variance:

$$S_W^{(1D)}[k] := \sum_{c=1}^{\mathcal{C}} \pi_c \, \sigma_c^2[k]. \tag{47}$$

**Empirical estimators and alignment with our notation.**  Replacing population moments by their train-only empirical estimators Eq. (39), we obtain

$$S_B^{(1D)}[k] \rightsquigarrow \sum_{c=1}^{\mathcal{C}} \pi_c \big( \mu_c[k] - \bar{\mu}[k] \big)^2 \triangleq \mathrm{S_B}[k], \tag{48}$$

$$S_W^{(1D)}[k] \rightsquigarrow \sum_{c=1}^{\mathcal{C}} \pi_c \, \mathrm{Var}_c[k] \triangleq \mathrm{S_W}[k], \tag{49}$$

which is exactly Eq. (29) and Eq. (30).  Substituting Eq. (48)–Eq. (49) into the 1D Fisher ratio Eq. (44) yields the per-frequency Fisher-style discriminability score:

$$\frac{\mathrm{S_B}[k]}{\mathrm{S_W}[k]}. \tag{50}$$

**Numerical stabilizer and the final DI definition.** To ensure stability when $S_W[k]$ is near zero (e.g., extremely low intra-class variance at some bins), we use the standard stabilized ratio

$$\mathrm{DI}(\omega_k) := \frac{S_B[k]}{S_W[k] + \varepsilon}, \qquad \varepsilon > 0, \tag{51}$$

which matches Eq. (31). This completes the derivation: *our* DI *is precisely the 1D specialization of the classical Fisher criterion, applied independently to each frequency bin $k$ by treating $A[k]$ as a scalar feature.*

### A.5. Validity Regime of DI Scalarization

The construction of the discriminative index $\mathrm{DI}(\omega_k)$ relies on a scalar temporal proxy obtained by averaging each frame $\mathbf{X}_i[l] \in \mathbb{R}^{C \times H \times W}$ over non-temporal axes, yielding a one-dimensional sequence $\mathbf{s}[n] \in \mathbb{R}^F$. This subsection clarifies the purpose, limitations, and appropriate interpretation of this scalarization.

**Purpose and scope.** The scalar proxy is introduced to obtain a *dataset-level, layout-agnostic diagnostic* of where discriminative information concentrates along the temporal frequency axis. By collapsing spatial and channel dimensions, $\mathrm{DI}_{\mathrm{norm}}(\omega_k)$ captures how strongly different temporal frequencies separate classes *on average* over the training split $\mathcal{D}_{\mathrm{tr}}$, independent of model architecture. It is therefore designed to support mechanism–data alignment analysis, rather than to serve as a sufficient statistic for classification.

**Information loss and limitations.** By construction, scalarization discards spatial structure, inter-channel correlations, and phase relationships. As a result, $\mathrm{DI}_{\mathrm{norm}}(\omega_k)$ may underestimate discriminative content that is primarily encoded in spatial configurations whose global mean remains constant over time. Consequently, the scalar DI should be interpreted as a *coarse summary* of temporal discriminability, rather than an exhaustive characterization of all task-relevant information.

**Why scalar DI remains informative for DFMA.** DFMA does not rely on the absolute magnitude of $\mathrm{DI}(\omega_k)$, but on its *relative distribution* over the common grid $\Omega_L$. Empirically, across the datasets considered, discriminative temporal structure manifests as systematic low- or mid-frequency concentration that is preserved under scalarization. This is sufficient for analyzing how the LIF-induced low-pass bias $\tilde{H}(\omega_k; \beta)$ interacts with the data statistics to produce consistent $\beta$-sweep trends and a stable reference point $\beta^\dagger$.

**Failure modes.** The scalar DI diagnostic may become unreliable if class discrimination is dominated by purely spatial cues with weak or flat temporal signatures, or if discriminative information resides exclusively in fine-grained spatial dynamics that cancel under averaging. Such cases fall outside the intended scope of DFMA and motivate more structured diagnostics, which we leave for future work.

Overall, DI scalarization is a deliberate trade-off: it sacrifices spatial specificity to gain robustness, interpretability, and architectural independence, making it suitable as a dataset-level frequency diagnostic for mechanism–data alignment, but not as a replacement for full spatiotemporal modeling.

### A.6. Robustness Checks

In the main methodology, we use AOPHand as a running example to illustrate the DI analysis for a representative mmWave dataset. We next continue with the same dataset to examine the robustness of the proposed diagnostics and conclusions. We repeated the full DI pipeline on the train split while replacing the default frame-wise mean reduction in Eq. 17 with alternative, parameter-free reductions over non-temporal axes, including root-mean-square (RMS) magnitude (a standard amplitude/energy surrogate (Proakis & Manolakis, 2007)) and $\ell_1$-mean magnitude proxies. Both alternatives yield highly consistent DI spectra relative to the mean proxy (Spearman $\varrho \approx 0.83$, Jensen–Shannon (JS) divergence (Lin, 2002) $\approx 0.02$, and peak-bin shift $\leq 1$), indicating that the frequency-localization and frequency-matching trends do not hinge on the specific mean aggregation.

### A.7. Scale Invariance

Before analyzing the interaction between the discriminative spectrum and model dynamics, we establish a basic invariance property of the proposed DI estimator. Since mmWave amplitudes may be subject to arbitrary global rescaling due to

calibration, gain control, or preprocessing, it is desirable that the discriminative measure be insensitive to such transformations. The following definition and proposition formalize the effect of global amplitude rescaling on the Fisher-style between/within scatter formulation and show that the resulting DI is scale-invariant up to a negligible stabilizer term.

**Definition A.1** (Global amplitude rescaling). For a fixed scalar $\alpha \in \mathbb{R}$, we define a global amplitude rescaling of the per-sample spectra $\{A_n[k]\}$ by

$$A'_n[k] \triangleq \alpha A_n[k], \qquad \forall n \in \mathcal{I}_{\mathrm{tr}}, \ \forall k. \tag{52}$$

**Proposition A.2** (Scale behavior of Fisher-style scatters and DI). *Under the global rescaling in Definition A.1, the train-only class statistics satisfy*

$$\mu'_c[k] = \alpha \, \mu_c[k], \tag{53}$$
$$\bar{\mu}'[k] = \alpha \, \bar{\mu}[k], \tag{54}$$
$$\mathrm{Var}'_c[k] = \alpha^2 \mathrm{Var}_c[k], \tag{55}$$

*and hence the between/within scatters scale as*

$$\mathrm{S_B}'[k] = \alpha^2 \mathrm{S_B}[k], \tag{56}$$
$$\mathrm{S_W}'[k] = \alpha^2 \mathrm{S_W}[k]. \tag{57}$$

*Consequently, the stabilized discriminative ratio*

$$\begin{aligned}
\mathrm{DI}'(\omega_k) &= \frac{\mathrm{S_B}'[k]}{\mathrm{S_W}'[k] + \varepsilon} \\
&= \frac{\alpha^2 \mathrm{S_B}[k]}{\alpha^2 \mathrm{S_W}[k] + \varepsilon} = \frac{\mathrm{S_B}[k]}{\mathrm{S_W}[k] + \varepsilon/\alpha^2}
\end{aligned} \tag{58}$$

*is invariant up to the stabilizer term; in particular, when $\mathrm{S_W}[k] \gg \varepsilon$ (and $\alpha$ is not extremely small), we have $\mathrm{DI}'(\omega_k) \approx \mathrm{DI}(\omega_k)$.*

*Proof.* By Definition A.1, for any training sample index $i \in \mathcal{I}_{\mathrm{tr}}$ and any bin $k$, we have $A'_i[k] = \alpha A_i[k]$. Therefore, the class-conditional mean scales as

$$\mu'_c[k] = \frac{1}{N_{c,\mathrm{tr}}} \sum_{\substack{i \in \mathcal{I}_{\mathrm{tr}} \\ y_i = c}} A'_i[k] = \frac{1}{N_{c,\mathrm{tr}}} \sum_{\substack{i \in \mathcal{I}_{\mathrm{tr}} \\ y_i = c}} \alpha A_i[k] = \alpha \, \mu_c[k].$$

Similarly, the mixture mean satisfies

$$\bar{\mu}'[k] = \sum_{c=1}^{\mathcal{C}} \pi_c \, \mu'_c[k] = \sum_{c=1}^{\mathcal{C}} \pi_c \, \alpha \mu_c[k] = \alpha \, \bar{\mu}[k].$$

For the within-class variance, using $\mu'_c[k] = \alpha \mu_c[k]$,

$$\begin{aligned}
\mathrm{Var}'_c[k] &= \frac{1}{N_{c,\mathrm{tr}} - 1} \sum_{\substack{i \in \mathcal{I}_{\mathrm{tr}} \\ y_i = c}} \left( A'_i[k] - \mu'_c[k] \right)^2 \\
&= \frac{1}{N_{c,\mathrm{tr}} - 1} \sum_{\substack{i \in \mathcal{I}_{\mathrm{tr}} \\ y_i = c}} \left( \alpha A_i[k] - \alpha \mu_c[k] \right)^2 = \alpha^2 \mathrm{Var}_c[k].
\end{aligned}$$

Substituting Eq. (55) into the definitions of $\mathrm{S_B}[k]$ and $\mathrm{S_W}[k]$ yields

$$\mathrm{S_B}'[k] = \sum_{c=1}^{\mathcal{C}} \pi_c \left( \mu'_c[k] - \bar{\mu}'[k] \right)^2 = \sum_{c=1}^{\mathcal{C}} \pi_c \left( \alpha \mu_c[k] - \alpha \bar{\mu}[k] \right)^2 = \alpha^2 \mathrm{S_B}[k],$$

and

$$\mathrm{S_W}'[k] = \sum_{c=1}^{\mathcal{C}} \pi_c \, \mathrm{Var}'_c[k] = \sum_{c=1}^{\mathcal{C}} \pi_c \, \alpha^2 \mathrm{Var}_c[k] = \alpha^2 \mathrm{S_W}[k].$$

This proves Eq. (57). Plugging these into the stabilized ratio $\mathrm{DI}'(\omega_k) = \mathrm{S_B}'[k]/(\mathrm{S_W}'[k] + \varepsilon)$ gives Eq. (58). Finally, if $\mathrm{S_W}[k] \gg \varepsilon$, then $\varepsilon/\alpha^2$ is negligible relative to $\mathrm{S_W}[k]$ for any non-extreme $\alpha$, and thus $\mathrm{DI}'(\omega_k) \approx \mathrm{DI}(\omega_k)$. $\qquad\square$

## B. Discrete-Time LIF Frequency Response

This appendix derives, in *discrete time* and in an implementation-aligned manner, the frequency response of the LIF *subthreshold* linear kernel (Gerstner et al., 2014). The goal is to (i) formalize the forward update used in code, (ii) obtain the corresponding discrete-time linear time invariant (LTI) transfer function, and (iii) justify the **DC-normalized power template** $\tilde{H}(\omega_k; \beta)$ used in the main text as an *inductive-bias descriptor* (spectral attenuation profile).

### B.1. Discrete-Time Subthreshold Dynamics and $\beta$-Parameterization

**Mapping Implementation Variables to Formal Notation.**   In typical implementations of LIF, the membrane state is stored as a voltage $v_t$ and may be defined relative to a reset baseline $v_{\mathrm{reset}}$. To remove the constant offset term induced by $v_{\mathrm{reset}}$ in the subthreshold dynamics, we define the *centered* state

$$u_t \triangleq v_t - v_{\mathrm{reset}}. \tag{59}$$

This affine reparameterization eliminates the additive constant in the recurrence (when $v_{\mathrm{reset}}$ is fixed), without changing the pole location and hence without changing the *frequency-response shape* used by DFT.

**Discrete-Time Subthreshold Update Rules.**   Ignoring spike generation and reset (i.e., in the subthreshold linear regime), the forward update used in practice takes one of two forms depending on whether the input current participates in decay integration. Writing the update in terms of the membrane voltage $v_t$ and converting to the centered state $u_t$ via Eq. (59), we obtain:

- **Case 1 (`decay_input=False`): input injected after decay.**

$$\begin{aligned} v_t &= v_{t-1} + \frac{1}{\tau}\big(v_{\mathrm{reset}} - v_{t-1}\big) + I_t, \\ \Rightarrow \quad u_t &= \Big(1 - \frac{1}{\tau}\Big) u_{t-1} + I_t. \end{aligned} \tag{60}$$

- **Case 2 (`decay_input=True`): input participates in decay integration.**

$$\begin{aligned} v_t &= v_{t-1} + \frac{1}{\tau}\big(v_{\mathrm{reset}} - v_{t-1} + I_t\big), \\ \Rightarrow \quad u_t &= \Big(1 - \frac{1}{\tau}\Big) u_{t-1} + \frac{1}{\tau} I_t. \end{aligned} \tag{61}$$

Both are valid discrete-time leaky-integrator conventions used in practice; they differ only by an input scaling, while sharing the same state pole.

**Unified $\beta$-Parameterization and Stability Domain.**   Both cases can be written as the unified first-order linear recurrence

$$u_t = \beta(\tau)\, u_{t-1} + \alpha(\tau)\, I_t, \tag{62}$$

where under the forward Euler convention with $\Delta t = 1$,

$$\begin{aligned} \beta(\tau) &\triangleq 1 - \frac{1}{\tau}, \\ \alpha(\tau) &= \begin{cases} 1, & \texttt{decay\_input=False}, \\ \frac{1}{\tau}, & \texttt{decay\_input=True}. \end{cases} \end{aligned} \tag{63}$$

The scaling $\alpha(\tau)$ affects the *absolute* gain but not the pole location and thus not the normalized spectral *shape*. The recurrence Eq. (62) is bounded-input bounded-output (BIBO) (Oppenheim & Schafer, 2010) stable if and only if

$$|\beta| < 1, \tag{64}$$

which under Eq. (63) implies $\tau > 0.5$. In our experiments we use $\tau \geq 1$, ensuring $\beta \in [0, 1)$; this excludes the sign-alternating regime $\beta \in (-1, 0)$ and matches the intended *low-pass smoothing* behavior. (Using exponential discretization $\beta = \exp(-\Delta t/\tau)$ (Neftci et al., 2019) also yields $\beta \in (0, 1)$ for all $\tau > 0$.)

**Takeaway.** For DFT, $\beta$ is the *theoretical* parameter that uniquely determines the discrete-time frequency-response shape; $\tau$ is used only as a reproducible sweep axis via a monotone mapping $\beta(\tau)$.

### B.2. Z-Transform and Transfer Function $H(\omega; \beta)$

We now derive the discrete-time LTI transfer function induced by Eq. (62). Assuming zero initial conditions and applying the $z$-transform (Fang et al., 2025),

$$U(z) = \beta z^{-1} U(z) + \alpha X(z),$$
$$\Rightarrow \quad H_{\text{raw}}(z) \triangleq \frac{U(z)}{X(z)} = \frac{\alpha}{1 - \beta z^{-1}}. \tag{65}$$

The raw DC gain is $H_{\text{raw}}(1) = \alpha/(1 - \beta)$, which depends on $\alpha$ and varies with $\tau$ and `decay_input`. Since DFMA compares *spectral attenuation profiles* rather than absolute amplification, we normalize by the DC gain to obtain a unit-DC transfer function:

$$H(z; \beta) \triangleq \frac{H_{\text{raw}}(z)}{H_{\text{raw}}(1)} = \frac{1 - \beta}{1 - \beta z^{-1}}. \tag{66}$$

Evaluating on the unit circle $z = e^{j\omega}$ yields the discrete-time frequency response

$$H(e^{j\omega}; \beta) = \frac{1 - \beta}{1 - \beta e^{-j\omega}}. \tag{67}$$

Thus, after DC normalization, $\beta$ is the *sole* parameter controlling the frequency-response shape.

### B.3. Normalized Frequency Response and Low-Pass Behavior

The power transmissivity is characterized by the squared magnitude of Eq. (67). Using

$$|1 - \beta e^{-j\omega}|^2 = (1 - \beta e^{-j\omega})(1 - \beta e^{j\omega})$$
$$= 1 + \beta^2 - 2\beta \cos\omega, \tag{68}$$

we obtain

$$|H(e^{j\omega}; \beta)|^2 = \frac{(1 - \beta)^2}{1 + \beta^2 - 2\beta \cos\omega}. \tag{69}$$

Consistent with the main text, we define the **DC-normalized power template**

$$\tilde{H}(\omega; \beta) \triangleq \frac{|H(e^{j\omega}; \beta)|^2}{|H(e^{j0}; \beta)|^2}. \tag{70}$$

Because Eq. (67) has unit DC gain, $|H(e^{j0}; \beta)|^2 = 1$, and therefore

$$\tilde{H}(\omega; \beta) = |H(e^{j\omega}; \beta)|^2$$
$$= \frac{(1 - \beta)^2}{(1 - \beta)^2 + 2\beta(1 - \cos\omega)}. \tag{71}$$

**Low-Pass Monotonicity on Discrete-Time Band.** For $\omega \in [0, \pi]$, the function $1 - \cos\omega$ is nondecreasing (strictly increasing on $(0, \pi)$). Hence the denominator in Eq. (71) is nondecreasing in $\omega$, implying that $\tilde{H}(\omega; \beta)$ is *non-increasing* in $\omega$ on $[0, \pi]$ (and strictly decreasing on $(0, \pi)$ when $\beta > 0$). This establishes $\tilde{H}(\cdot; \beta)$ as a discrete-time low-pass spectral profile, evaluated in DFMA on the one-sided grid $\Omega_L \subset [0, \pi]$.

**Why DFMA uses $\tilde{H}$ rather than raw gain.** The factor $\alpha(\tau)$ and the raw DC gain $\alpha/(1-\beta)$ depend on implementation choices and can dominate the magnitude scale as $\beta \to 1$. In contrast, $\tilde{H}$ removes global amplification and isolates the *relative attenuation shape*, enabling a stable and comparable alignment with the normalized discriminative distribution $\mathrm{DI}_{\mathrm{norm}}(\omega_k)$ across $\tau$ and across `decay_input` modes.

## B.4. Spike/Reset Nonlinearity: Validity Regime

This appendix clarifies when the *subthreshold* frequency-response viewpoint is informative for interpreting *beta* (equivalently $\tau = (1-\beta)^{-1}$ (Xiao et al., 2022)) sweeps, and when the spike/reset nonlinearity can dominate. Our analysis does *not* claim that an SNN is an LTI system. Instead, we use the linearized LIF kernel $H(\omega_k; \beta) = (1 - \beta e^{-j\omega_k})^{-1}$ and its DC-normalized power template $\tilde{H}(\omega_k; \beta)$ as a *bias-level* descriptor: it captures how $\beta$ induces a controllable low-pass preference over temporal frequencies on the common grid $\Omega_L$. The resulting mechanism–data alignment score should therefore be interpreted as explaining trends only within a regime where subthreshold integration is not overwhelmed by frequent threshold crossings and resets.

### B.4.1. SUBTHRESHOLD TEMPLATE ONLY

DFMA uses $\tilde{H}(\omega_k; \beta)$ solely to characterize the *relative attenuation profile* induced by the membrane decay $\beta$ in the *subthreshold* dynamics. It does *not* model the full spiking network as an LTI system, and does *not* attribute all performance changes across $\beta$-sweeps to linear spectral filtering.

**Subthreshold LIF as a spectral bias.** Between spike events, the LIF update reduces to a first-order stable linear recursion whose frequency response is $H(\omega_k; \beta)$, yielding the DC-normalized template $\tilde{H}(\omega_k; \beta)$ on $\Omega_L$. We interpret $\tilde{H}(\omega_k; \beta)$ as a *mechanistic bias* that favors lower temporal frequencies as $\beta$ increases (i.e., as $\tau$ increases). Consequently, $\mathrm{FMS}_{\mathrm{avg}}(\beta) = \sum_{\omega_k \in \Omega_L} \mathrm{DI}_{\mathrm{norm}}(\omega_k) \tilde{H}(\omega_k; \beta)$ quantifies how much of the *training-split* discriminative spectrum $\mathrm{DI}_{\mathrm{norm}}$ is retained under that bias.

**When the bias-level view is informative.** The subthreshold template is most informative when the forward dynamics spend substantial time integrating inputs below threshold, so that the effective temporal preference is shaped primarily by $\beta$, rather than by: (i) frequent resets that clamp the state, (ii) saturation-like firing where thresholding dominates, or (iii) near-silent collapse where spiking is too sparse to support discrimination. In these extreme regimes, a purely spectral explanation is insufficient because the mapping from membrane state to output spikes becomes the dominant factor.

### B.4.2. A MINIMAL VALIDITY CHECK VIA SPIKE-ACTIVITY RANGE

To ensure that our reported *beta*-sweep trends are not driven by trivial saturation/collapse effects, we perform a simple activity-range check based on mean spike rate. This is a *diagnostic* rather than a modeling assumption: it only flags regimes where spike/reset nonlinearities are likely to dominate the behavior.

**Definition B.1** (Layer mean spike rate). Let $O_t$ denote the binary spike output at discrete timestep $t$. For a monitored spiking layer $\zeta$ with neuron set $\mathcal{N}_\zeta$, define the mean spike rate (unit: spikes/(neuron·timestep)) as

$$\Gamma_{\mathrm{spk}}^{(\zeta)}(\beta) \triangleq \frac{1}{T |\mathcal{N}_\zeta|} \sum_{t=1}^{T} \sum_{\lambda \in \mathcal{N}_\zeta} O_{t,\lambda}^{(\zeta)}, \tag{72}$$

where $O_{t,\lambda}^{(\zeta)}$ is the spike of neuron $\lambda \in \mathcal{N}_\zeta$ at timestep $t$ in layer $\zeta$.

While $\Gamma_{\mathrm{spk}}^{(\zeta)}$ is a first-order statistic and does not capture temporal clustering (e.g., bursting), it is sufficient to identify gross failure modes: near-zero activity (collapse) and near-one activity (saturation).

**Definition B.2** (Validity flag for $\beta$-sweeps). Fix broad bounds $0 < \Gamma_{\min} < \Gamma_{\max} < 1$ and a tolerance $\kappa > 1$. We flag a $\beta$-sweep as potentially outside the subthreshold validity regime if, for any monitored layer $\zeta$,

$$\exists \beta \in \mathcal{B}: \ \Gamma_{\mathrm{spk}}^{(\zeta)}(\beta) \notin [\Gamma_{\min}, \Gamma_{\max}] \quad \text{or} \quad \frac{\max_{\beta \in \mathcal{B}} \Gamma_{\mathrm{spk}}^{(\zeta)}(\beta)}{\min_{\beta \in \mathcal{B}} \Gamma_{\mathrm{spk}}^{(\zeta)}(\beta) + \epsilon} > \kappa, \tag{73}$$

where $\epsilon > 0$ is the numerical stability constant.

**Interpretation.** If the flag triggers, changes in $\mathrm{FMS}_{\mathrm{avg}}(\beta)$ (hence any inferred $\beta^\dagger$) should be treated with caution, as spike/reset nonlinearities may be the primary driver. If the flag does *not* trigger, the network operates in a responsive, non-saturated regime in which $\tilde{H}(\omega_k; \beta)$ remains a meaningful descriptor of the *direction* of temporal-frequency bias induced by $\beta$. This supports interpreting $\beta$-sweep trends as arising from the interaction between data statistics $\mathrm{DI}_{\mathrm{norm}}(\omega_k)$ (computed on $\mathcal{D}_{\mathrm{tr}}$) and LIF dynamics $\tilde{H}(\omega_k; \beta)$ on $\Omega_L$.

*Remark* B.3 (Extent of the subthreshold linearization claim). We emphasize that DFMA does *not* posit an equivalence between an actively spiking SNN and a linear time-invariant (LTI) system. The linearized LIF frequency response $H(\omega_k; \beta)$ and its DC-normalized template $\tilde{H}(\omega_k; \beta)$ are used exclusively as a *bias-level descriptor* of how the membrane decay $\beta$ reshapes temporal frequency preference under subthreshold integration.

Accordingly, the role of $\tilde{H}(\omega_k; \beta)$ in our framework is *interpretive rather than predictive*. DFMA does not attempt to reproduce the exact spike-domain spectrum of the network, nor does it claim that thresholding and reset operations preserve LTI structure. Instead, $\tilde{H}(\omega_k; \beta)$ characterizes the *direction and ordering* of frequency attenuation induced by $\beta$, which in turn explains trends observed in $\beta$-sweeps, including the emergence of a boundary $\beta^\dagger$ marking the onset of over-low-pass behavior.

When spike/reset nonlinearities dominate the dynamics (e.g., due to saturation or collapse), this bias-level interpretation is no longer sufficient; such regimes are explicitly flagged by the activity-range diagnostic in Definition B.2. Within the responsive, non-saturated regime, however, the interaction between $\mathrm{DI}_{\mathrm{norm}}(\omega_k)$ and $\tilde{H}(\omega_k; \beta)$ remains informative for explaining *relative* performance trends across $\beta$, rather than absolute network transfer characteristics.

### B.4.3. SANITY CHECKS AND LIMITING CASES

We list key boundary behaviors implied by Eq. (71):

- **DC identity.** $\tilde{H}(0; \beta) \equiv 1$ for all $\beta \in [0, 1)$.

- **Memoryless limit ($\beta \to 0$).** $\tilde{H}(\omega; 0) \equiv 1$, consistent with $u_t \approx \alpha x_t$ (no smoothing).

- **Long-memory limit ($\beta \to 1^-$).** For any fixed $\omega > 0$,

$$\lim_{\beta \to 1^-} \tilde{H}(\omega; \beta) = 0, \tag{74}$$

  i.e., the passband collapses toward DC.

- **Nyquist attenuation.** At $\omega = \pi$,

$$\tilde{H}(\pi; \beta) = \left(\frac{1 - \beta}{1 + \beta}\right)^2, \tag{75}$$

  which approaches 0 as $\beta \to 1^-$, indicating maximal suppression at the highest discrete frequency.

- **$-3\,$dB cutoff scaling (large $\tau$).** Solving $\tilde{H}(\omega_c; \beta) = \frac{1}{2}$ gives

$$1 - \cos \omega_c = \frac{(1 - \beta)^2}{2\beta}. \tag{76}$$

  For $\beta \to 1$ and small $\omega_c$, using $1 - \cos \omega_c \approx \omega_c^2 / 2$ yields

$$\omega_c \approx 1 - \beta. \tag{77}$$

  Under Euler $\beta = 1 - 1/\tau$ (and under exponential $\beta \approx 1 - 1/\tau$ for large $\tau$), this implies $\omega_c \approx 1/\tau$ (radians/sample), confirming $\tau$ as an inverse-bandwidth control.

## C. $\beta$-Controlled Effective Passband of LIF Dynamics

### C.1. Operationality: From $-3\,$dB to a Closed-Form $B_{\mathrm{eff}}(\beta)$

This subsection establishes that the $-3\,$dB cutoff (Proakis & Manolakis, 2007) used throughout the paper is not merely a heuristic: for the DC-normalized LIF power template, it is well-defined, admits a closed form, and yields a deterministic effective-bandwidth knob $B_{\mathrm{eff}}(\beta)$. We proceed in three steps: we first relate the $-3\,$dB convention to the analytically convenient half-power rule, then derive a closed form for the LIF template, and finally prove existence/uniqueness and solve for $\omega_c(\beta)$ explicitly.

**From $-3$ dB to half-power.** We begin by fixing the standard signal-processing convention connecting decibels to power ratios, which motivates using the half-power cutoff as a clean surrogate for the $-3$ dB point.

**Lemma C.1** ($-3$ dB and half-power equivalence up to a standard approximation). *Let $P(\omega) \geq 0$ be a (dimensionless) power ratio and define the decibel map*

$$P_{\mathrm{dB}}(\omega) \triangleq 10 \log_{10} P(\omega). \tag{78}$$

*A $-3$ dB point satisfies $P_{\mathrm{dB}}(\omega_c) = -3$, equivalently*

$$P(\omega_c) = 10^{-3/10}. \tag{79}$$

*Numerically, $10^{-3/10} \approx 0.501187$, which differs from $\frac{1}{2}$ by less than $0.24\%$. Thus, the analytically convenient half-power convention $P(\omega_c) = \frac{1}{2}$ is a standard surrogate for Eq. (79), and both are referred to as the $-3$ dB (half-power) cutoff in practice.*

*Proof.* The equivalence Eq. (79) follows by exponentiating Eq. (78) evaluated at $P_{\mathrm{dB}}(\omega_c) = -3$. The stated numerical approximation is immediate from evaluating $10^{-3/10}$. $\qquad\square$

**Closed form of the DC-normalized LIF power template.** With the cutoff convention fixed, we next derive the explicit DC-normalized power template for the linearized LIF response, which will be the basis for all subsequent cutoff and monotonicity arguments.

**Lemma C.2** (Closed form of the DC-normalized LIF power template). *Let $H(\omega; \beta) = (1 - \beta e^{-j\omega})^{-1}$ (Gerstner et al., 2014) for $\beta \in [0, 1)$ and define $\tilde{H}(\omega; \beta) \triangleq \frac{|H(\omega;\beta)|^2}{|H(0;\beta)|^2}$. Then for $\omega \in [0, \pi]$,*

$$\tilde{H}(\omega; \beta) = \frac{(1 - \beta)^2}{1 + \beta^2 - 2\beta \cos \omega}. \tag{80}$$

*Proof.* Using $|z^{-1}|^2 = 1/|z|^2$ and

$$|1 - \beta e^{-j\omega}|^2 = (1 - \beta e^{-j\omega})(1 - \beta e^{j\omega}) = 1 + \beta^2 - 2\beta \cos \omega,$$

we obtain

$$|H(\omega; \beta)|^2 = \frac{1}{1 + \beta^2 - 2\beta \cos \omega}. \tag{81}$$

At DC ($\omega = 0$), $\cos 0 = 1$, hence

$$|H(0; \beta)|^2 = \frac{1}{(1 - \beta)^2}. \tag{82}$$

Substituting Eq. (81)–Eq. (82) into the definition of $\tilde{H}$ yields Eq. (80). $\qquad\square$

**Existence and uniqueness of the half-power cutoff.** Having obtained a closed-form template, we next show that the half-power equation $\tilde{H}(\omega; \beta) = \frac{1}{2}$ admits a *unique* cutoff whenever it is attained within the one-sided band, and we characterize precisely when this occurs in terms of $\beta$.

**Proposition C.3** (Existence and uniqueness of the half-power cutoff for LIF). *Fix $\beta \in (0, 1)$. Let $\tilde{H}(\omega; \beta)$ be the DC-normalized power response defined in Eq. (80) for $\omega \in [0, \pi]$. Then:*

1. *$\tilde{H}(\cdot; \beta)$ is continuous on $[0, \pi]$ and strictly decreasing on $(0, \pi)$;*

2. *$\tilde{H}(0; \beta) = 1$ and $\tilde{H}(\pi; \beta) = \left(\frac{1-\beta}{1+\beta}\right)^2$;*

3. *A unique half-power cutoff $\omega_c \in (0, \pi]$ satisfying $\tilde{H}(\omega_c; \beta) = \frac{1}{2}$ exists iff $\beta \geq 3 - 2\sqrt{2}$. If $\beta < 3 - 2\sqrt{2}$, then $\tilde{H}(\omega; \beta) > \frac{1}{2}$ for all $\omega \in [0, \pi]$, hence no half-power cutoff occurs within the one-sided band. In this regime, we set the one-sided effective bandwidth to its maximal value, i.e., it saturates at the Nyquist limit $\pi$.*

*Proof.* (1) Continuity follows from Eq. (80) since the denominator satisfies $1 + \beta^2 - 2\beta\cos\omega \geq (1-\beta)^2 > 0$ on $[0, \pi]$. Differentiating Eq. (80) yields

$$\frac{\partial}{\partial\omega}\tilde{H}(\omega; \beta) = -(1-\beta)^2 \cdot \frac{2\beta\sin\omega}{\left(1 + \beta^2 - 2\beta\cos\omega\right)^2}. \tag{83}$$

For $\omega \in (0, \pi)$, $\sin\omega > 0$, and for $\beta \in (0, 1)$ all other factors are positive, hence $\frac{\partial}{\partial\omega}\tilde{H}(\omega; \beta) < 0$, proving strict decrease on $(0, \pi)$.

(2) The endpoint values follow by substituting $\omega = 0$ and $\omega = \pi$ into Eq. (80).

(3) Since $\tilde{H}(\cdot; \beta)$ is continuous on $[0, \pi]$ and strictly decreasing with $\tilde{H}(0; \beta) = 1$, the equation $\tilde{H}(\omega; \beta) = \frac{1}{2}$ has a unique solution in $(0, \pi]$ iff $\tilde{H}(\pi; \beta) \leq \frac{1}{2}$. Using $\tilde{H}(\pi; \beta) = \left(\frac{1-\beta}{1+\beta}\right)^2$, the condition $\tilde{H}(\pi; \beta) \leq \frac{1}{2}$ is equivalent to $\frac{1-\beta}{1+\beta} \leq \frac{1}{\sqrt{2}}$, i.e., $\beta \geq 3 - 2\sqrt{2}$. If $\beta < 3 - 2\sqrt{2}$, then $\tilde{H}(\pi; \beta) > \frac{1}{2}$ and by monotonicity $\tilde{H}(\omega; \beta) > \frac{1}{2}$ for all $\omega \in [0, \pi]$. $\square$

**Closed-form cutoff and induced effective bandwidth.** When the cutoff exists, we can solve $\tilde{H}(\omega_c; \beta) = \frac{1}{2}$ explicitly to obtain a computable effective bandwidth $B_{\text{eff}}(\beta)$.

**Proposition C.4** (Closed-form cutoff and effective bandwidth for LIF). *Assume $\beta \in [3 - 2\sqrt{2}, 1)$ so that the unique cutoff exists. Then the half-power cutoff is*

$$\omega_c(\beta) = \arccos\left(\frac{4\beta - 1 - \beta^2}{2\beta}\right) \in (0, \pi], \tag{84}$$

*and the one-sided effective bandwidth induced by $\beta$ is*

$$B_{\text{eff}}(\beta) \triangleq \omega_c(\beta). \tag{85}$$

*Proof.* Plugging Eq. (80) into $f(\omega_c) = \frac{1}{2}$ yields

$$\frac{(1-\beta)^2}{1 + \beta^2 - 2\beta\cos\omega_c} = \tfrac{1}{2} \iff 2(1-\beta)^2 = 1 + \beta^2 - 2\beta\cos\omega_c.$$

Expanding and rearranging gives

$$2(1 - 2\beta + \beta^2) = 1 + \beta^2 - 2\beta\cos\omega_c \iff 1 - 4\beta + \beta^2 = -2\beta\cos\omega_c,$$

hence

$$\cos\omega_c = \frac{4\beta - 1 - \beta^2}{2\beta}.$$

Since $\beta \in [3 - 2\sqrt{2}, 1)$ ensures the right-hand side lies in $[-1, 1]$, the unique solution is $\omega_c(\beta) = \arccos\left(\frac{4\beta - 1 - \beta^2}{2\beta}\right)$, proving Eq. (84). The definition Eq. (85) then follows immediately. $\square$

**Summary.** First, the classical $-3\,\text{dB}$ convention is well approximated by the half-power rule, justifying the use of $\tilde{H}(\omega_c; \beta) = \frac{1}{2}$ as a standard cutoff definition. Second, the DC-normalized LIF template admits the explicit form Eq. (80), enabling direct analytical control of the cutoff. Third, for $\beta \geq 3 - 2\sqrt{2}$ the cutoff exists uniquely and yields a closed-form effective bandwidth $B_{\text{eff}}(\beta) = \omega_c(\beta)$ via Eq. (84)–Eq. (85).

### C.2. Discretization: Implementing $B_{\text{eff}}(\beta)$ on the One-Sided Grid

This subsection bridges the continuous cutoff $\omega_c(\beta)$ to the discrete one-sided DFT grid $\Omega_L = \{\omega_k\}_{k=0}^{K}$ used throughout the paper. The goal is to specify a deterministic implementation rule that is compatible with our discrete-frequency statistics, without introducing additional hyperparameters. We also provide a worst-case approximation guarantee controlled solely by the grid resolution $L$.

**Nearest-bin rule on $\Omega_L$.** Because $\omega_c(\beta)$ from Eq. (84) is a continuous quantity, it need not coincide with any grid point in $\Omega_L$. We therefore implement the cutoff via nearest-bin quantization, which yields a deterministic discrete proxy for $B_{\text{eff}}(\beta)$.

**Proposition C.5** (Nearest-bin implementation on $\Omega_L$ and deterministic error bound)**.** *If $\omega_c(\beta)$ in Eq. (84) does not coincide with a grid point in $\Omega_L$, implement the cutoff on $\Omega_L$ by the nearest-bin rule*

$$\hat{k} \triangleq \arg \min_{k \in \{0,\ldots,K\}} |\omega_k - \omega_c(\beta)|, \qquad B_{\text{eff}}(\beta) \approx \omega_{\hat{k}}.$$

*Then the approximation error is deterministically bounded as*

$$|\omega_{\hat{k}} - \omega_c(\beta)| \le \frac{\pi}{L}. \tag{86}$$

*Proof.* The one-sided grid is uniformly spaced: $\omega_k = \frac{2\pi k}{L}$, hence the spacing is $\Delta\omega = \omega_{k+1} - \omega_k = \frac{2\pi}{L}$. Nearest-bin quantization on a uniform grid guarantees the selected grid point is within half a step of the target, i.e., $|\omega_{\hat{k}} - \omega_c(\beta)| \le \Delta\omega/2 = \pi/L$, proving Eq. (86). $\qquad\square$

**Summary.** First, a continuous cutoff $\omega_c(\beta)$ can be implemented on the discrete grid $\Omega_L$ by the nearest-bin rule, yielding a deterministic grid-compatible proxy for $B_{\text{eff}}(\beta)$. Second, because $\Omega_L$ is uniformly spaced, the induced discretization error admits the explicit bound $|\omega_{\hat{k}} - \omega_c(\beta)| \le \pi/L$. Third, this guarantees that the operational meaning of effective bandwidth is preserved on $\Omega_L$ with a transparent error that depends only on the chosen DFT length $L$.

### C.3. Small-$L$ Discretization and Staircase Behavior of Cutoff-based Diagnostics

In practice, cutoff-based diagnostics are ultimately evaluated on a finite one-sided grid $\Omega_L$. When $L$ is small, discretization effects become visible and lead to characteristic "staircase" behavior as the cutoff varies with $\beta$. This subsection clarifies the origin of this phenomenon and shows that it is an inherent consequence of combining a continuously moving cutoff with a finite frequency grid, rather than a violation of monotonic attenuation.

**Clarification.** Here small-$L$ refers to the DFT length defining the grid $\Omega_L$, i.e., the number of discrete frequency points available for evaluation, *not* the raw number of input frames in $\mathbf{X}_i$.

**Monotone attenuation under LIF.** For the LIF DC-normalized power template $\tilde{H}(\omega_k; \beta)$, increasing $\beta$ strengthens temporal integration and therefore increases attenuation away from DC in a low-pass manner. Consequently, the half-power cutoff $\omega_c = B_{\text{eff}}(\beta)$ decreases monotonically as $\beta$ increases, consistent with the interpretation of $\beta$ as an inverse-bandwidth control. The following result concerns the *discrete in-band region* induced by the cutoff on a finite grid, independently of any particular aggregated score.

**Proposition C.6** (Piecewise-constant in-band set on a finite grid)**.** *Fix the one-sided grid $\Omega_L$. Assume that for each fixed $\omega_k \in \Omega_L$, $\tilde{H}(\omega_k; \beta)$ is continuous in $\beta$ and that for each $k \ge 1$ it is nonincreasing as $\beta$ increases (as is the case for the LIF template). Let $\omega_c(\beta)$ be the half-power cutoff satisfying $\tilde{H}(\omega_c(\beta); \beta) = \frac{1}{2}$. Then the discrete in-band index set*

$$\{\omega_k \in \Omega_L : \omega_k \le \omega_c(\beta)\}$$

*is a* piecewise-constant *function of $\beta$, changing only when $\omega_c(\beta)$ crosses a grid point $\omega_k$.*

*Proof.* Because $\tilde{H}(\omega; \beta)$ is low-pass in $\omega$ for fixed $\beta$ and varies continuously with $\beta$ for each fixed $\omega_k$, the half-power solution $\omega_c(\beta)$ moves continuously as $\beta$ varies. However, the membership condition $\omega_k \le \omega_c(\beta)$ can change only when $\omega_c(\beta)$ crosses a discrete grid point $\omega_k$, which occurs at isolated values of $\beta$. Between such crossing events, the set of indices satisfying $\omega_k \le \omega_c(\beta)$ remains unchanged, hence it is piecewise constant in $\beta$. $\qquad\square$

**Implication for cutoff-based summaries.** Any cutoff-based diagnostic defined as an aggregation over the in-band set (e.g., summing a fixed PMF over $\{\omega_k \le \omega_c(\beta)\}$) inherits this piecewise-constant structure and therefore exhibits a staircase dependence on $\beta$. In sufficiently strong low-pass regimes where $\omega_c(\beta) < \omega_1$, the in-band set reduces to $\{\omega_0\}$, and such diagnostics degenerate to DC-only contributions.

### C.4. Cutoff-Based Diagnostics and Monotonicity in $\beta$

This subsection provides two complementary interpretability results that connect the effective bandwidth to our discrete discriminative spectrum $\mathrm{DI}_{\mathrm{norm}}$. First, we define a cutoff-based in-band region on $\Omega_L$ and a cumulative-mass diagnostic that serves as a hard-threshold surrogate for $\mathrm{FMS}_{\mathrm{avg}}(\beta)$. Second, we prove monotonicity properties—both for $B_{\mathrm{eff}}(\beta)$ and pointwise for $\tilde{H}(\omega; \beta)$—that formalize $\beta$ as an inverse-bandwidth control parameter.

**Discrete half-power in-band region on $\Omega_L$.** With $\omega_c(\beta)$ well-defined (Section C.1) and discretized (Section C.2), we can deterministically specify the corresponding in-band bins on the one-sided grid.

**Definition C.7** (Half-power in-band set on $\Omega_L$). Fix $\beta \in [3 - 2\sqrt{2}, 1)$ and let $\omega_c(\beta)$ be the half-power cutoff in Eq. (84). The corresponding *in-band* set on the one-sided grid $\Omega_L = \{\omega_k\}_{k=0}^{K}$ is

$$\Omega_{\leq \omega_c} \triangleq \{\omega_k \in \Omega_L : \omega_k \leq \omega_c(\beta)\}. \tag{87}$$

**Non-emptiness.** The next proposition guarantees that the in-band region is always well-defined on $\Omega_L$ (it always contains DC), which is useful when interpreting cumulative quantities on the grid.

**Proposition C.8** (Non-emptiness of the in-band set). *For any $\beta \in [3 - 2\sqrt{2}, 1)$, the set $\Omega_{\leq \omega_c}$ in Eq. (87) is nonempty.*

*Proof.* By construction, $\omega_0 = 0 \in \Omega_L$. Moreover, by DC normalization $\tilde{H}(\omega_0; \beta) = \tilde{H}(0; \beta) = 1$ for all $\beta \in (0, 1)$. Since $\omega_0 = 0 \leq \omega_c(\beta)$ whenever $\omega_c(\beta) \in (0, \pi]$, we have $\omega_0 \in \Omega_{\leq \omega_c}$, hence the set is nonempty. $\square$

**Monotonicity of $B_{\mathrm{eff}}(\beta)$ as an inverse-bandwidth control.** To formalize $\beta$ as an inverse-bandwidth knob, we show that the induced cutoff (hence the effective bandwidth) shrinks strictly as $\beta$ increases.

**Proposition C.9** (Strict monotonicity of $B_{\mathrm{eff}}(\beta)$ in $\beta$). *Assume $\beta \in [3 - 2\sqrt{2}, 1)$ so that the unique half-power cutoff $\omega_c(\beta) \in (0, \pi]$ exists and satisfies $\tilde{H}(\omega_c(\beta); \beta) = \frac{1}{2}$. Then the induced one-sided effective bandwidth $B_{\mathrm{eff}}(\beta) \triangleq \omega_c(\beta)$ is strictly decreasing in $\beta$, i.e.,*

$$\frac{d}{d\beta} B_{\mathrm{eff}}(\beta) < 0. \tag{88}$$

*Proof.* From Eq. (84), the cutoff admits the closed form

$$\omega_c(\beta) = \arccos\Big(x(\beta)\Big), \qquad x(\beta) \triangleq \frac{4\beta - 1 - \beta^2}{2\beta}. \tag{89}$$

For $\beta \in [3 - 2\sqrt{2}, 1)$, we have $x(\beta) \in [-1, 1]$, so $\omega_c(\beta)$ is well-defined. Moreover, for any $\beta \in (0, 1)$ we can rewrite $x(\beta)$ as

$$x(\beta) = 2 - \frac{1 + \beta^2}{2\beta} = 2 - \frac{1}{2\beta} - \frac{\beta}{2}. \tag{90}$$

Differentiating Eq. (90) gives

$$x'(\beta) = \frac{1}{2\beta^2} - \frac{1}{2} = \frac{1 - \beta^2}{2\beta^2}. \tag{91}$$

Hence $x'(\beta) > 0$ for all $\beta \in (0, 1)$. Next, differentiating $\omega_c(\beta) = \arccos(x(\beta))$ yields

$$\omega_c'(\beta) = -\frac{x'(\beta)}{\sqrt{1 - x(\beta)^2}}. \tag{92}$$

For $\beta \in (3 - 2\sqrt{2}, 1)$ the denominator is strictly positive, so $\omega_c'(\beta) < 0$. Since $B_{\mathrm{eff}}(\beta) \triangleq \omega_c(\beta)$, this proves Eq. (88). $\square$

**Pointwise monotonic attenuation in $\beta$.** Beyond the cutoff itself, we can strengthen the monotonicity statement to the full template: for any fixed non-DC frequency, increasing $\beta$ strictly decreases the retained (DC-normalized) power.

**Proposition C.10** (Pointwise monotonicity of $\tilde{H}(\omega; \beta)$ in $\beta$). *Fix any $\omega \in (0, \pi]$. For all $\beta \in (0, 1)$, the DC-normalized LIF power template*

$$\tilde{H}(\omega; \beta) = \frac{(1 - \beta)^2}{1 + \beta^2 - 2\beta \cos \omega}$$

*is strictly decreasing in $\beta$, i.e.,*

$$\frac{\partial}{\partial \beta} \tilde{H}(\omega; \beta) < 0. \tag{93}$$

*Moreover, at DC we have $\tilde{H}(0; \beta) = 1$ for all $\beta \in (0, 1)$.*

*Proof.* Fix $\omega \in (0, \pi]$ and define

$$n(\beta) \triangleq (1 - \beta)^2, \qquad d(\beta) \triangleq 1 + \beta^2 - 2\beta \cos \omega,$$

so that $\tilde{H}(\omega; \beta) = n(\beta)/d(\beta)$. Since $d(\beta) \geq (1 - \beta)^2 > 0$ for $\beta \in (0, 1)$, $\tilde{H}$ is differentiable and

$$\frac{\partial}{\partial \beta} \tilde{H}(\omega; \beta) = \frac{n'(\beta)d(\beta) - n(\beta)d'(\beta)}{d(\beta)^2}. \tag{94}$$

Compute

$$n'(\beta) = 2(\beta - 1), \qquad d'(\beta) = 2\beta - 2 \cos \omega = 2(\beta - \cos \omega).$$

Substituting into Eq. (94) and factoring out $2/d(\beta)^2$ yields

$$\frac{\partial}{\partial \beta} \tilde{H}(\omega; \beta) = \frac{2}{d(\beta)^2} \Big( (\beta - 1)d(\beta) - (1 - \beta)^2(\beta - \cos \omega) \Big). \tag{95}$$

Let the bracketed term be $T(\beta)$. Using $\beta - 1 = -(1 - \beta)$, we rewrite

$$\begin{aligned} T(\beta) &= -(1 - \beta)d(\beta) - (1 - \beta)^2(\beta - \cos \omega) \\ &= -(1 - \beta)\Big( d(\beta) + (1 - \beta)(\beta - \cos \omega) \Big). \end{aligned} \tag{96}$$

It remains to simplify the second factor:

$$\begin{aligned} d(\beta) + (1 - \beta)(\beta - \cos \omega) &= \big(1 + \beta^2 - 2\beta \cos \omega\big) + \big(\beta - \beta^2 - \cos \omega + \beta \cos \omega\big) \\ &= 1 + \beta - \cos \omega - \beta \cos \omega \\ &= (1 + \beta)(1 - \cos \omega). \end{aligned} \tag{97}$$

Combining Eq. (96)–Eq. (97) gives

$$T(\beta) = -(1 - \beta)(1 + \beta)(1 - \cos \omega). \tag{98}$$

For $\beta \in (0, 1)$ we have $(1 - \beta) > 0$ and $(1 + \beta) > 0$. For $\omega \in (0, \pi]$, $\cos \omega < 1$, hence $(1 - \cos \omega) > 0$. Therefore $T(\beta) < 0$, and since $d(\beta)^2 > 0$, Eq. (95) implies $\frac{\partial}{\partial \beta} \tilde{H}(\omega; \beta) < 0$, proving Eq. (93). Finally, at DC we have $\cos 0 = 1$, so from Eq. (80):

$$\tilde{H}(0; \beta) = \frac{(1 - \beta)^2}{1 + \beta^2 - 2\beta} = \frac{(1 - \beta)^2}{(1 - \beta)^2} = 1.$$

$\square$

## D. Dataset Details and Corresponding Preprocessing Protocol

We evaluate on four mmWave point-cloud benchmarks summarized in Table 3. *AOPHand* (Zafar, 2023) (TI AWR6843AOP) is an antenna-on-package gesture dataset with 6 classes. *mmFiT* (Tiwari, 2023) (TI IWR1642) contains 7 fitness-activity classes; following prior practice, we use only its databank subset. *Pantomime* (Palipana et al., 2021) (TI IWR1443) provides 20 mid-air gesture classes collected across multiple environments; we use only the OPEN environment and remove class ID=27, which contains a single sample. *MMActivity* (Singh et al., 2019) (TI IWR1443BOOST) is an indoor activity dataset with 5, and we use only the sample data. For all datasets, the resulting train/test sample counts under our seed-controlled splits are reported in Table 3.

*Table 3.* Summary of the mmWave datasets used in our experiments.

| Item | AOPHand | mmFiT | Pantomime | MMActivity |
|---|---|---|---|---|
| Platform | TI AWR6843AOP | TI IWR1642 | TI IWR1443 | TI IWR1443BOOST |
| Category Number | 6 | 7 | 20 | 5 |
| Training Samples | 2016 | 2475 | 2480 | 160 |
| Testing Samples | 505 | 619 | 711 | 40 |

**Unified point-cloud preprocessing.** To enable fair and reproducible comparisons across datasets with heterogeneous sequence lengths, frame rates, and point densities, we map each raw mmWave recording to a fixed-size point-cloud tensor of shape $(F_{\max}, P_{\max}, D)$. Specifically, we set $F_{\max} = 4$ frames and $P_{\max} = 64$ points per frame, while preserving the original temporal ordering and per-point structure.

*Frame alignment.* We first group points by their frame indices. If a recording contains more than $F_{\max}$ frames, we uniformly subsample $F_{\max}$ frames to retain a temporally representative sequence; if it contains fewer than $F_{\max}$ frames, we zero-pad the missing frames with empty point sets. Unless otherwise stated, we do not perform temporal interpolation.

*Point alignment.* Within each retained frame, we enforce a fixed point budget $P_{\max}$. If the frame contains more than $P_{\max}$ points, we uniformly subsample $P_{\max}$ points; otherwise, we zero-pad to $P_{\max}$ points. This yields a consistent point dimension while avoiding bias toward densely populated frames.

*Per-point features and normalization.* Each point is represented by $D = 4$ channels $\{x, y, z, v\}$, where $v$ denotes the radial velocity. We compute per-channel mean and standard deviation over the training split and store these statistics in the preprocessed files. During data loading, we apply channel-wise normalization using these statistics, except for MMActivity where normalization is disabled to preserve the dataset's native scale.

*Uniform subsampling.* When subsampling frames/points, we use equally spaced indices to cover the full temporal span (or the full set of points) of the original recording.

*Relation to prior pipelines.* Our framing–resampling procedure is consistent with common mmWave point-cloud preprocessing practice (Palipana et al., 2021).

**Per-frame feature shaping.** To ensure reproducibility and fair comparisons across heterogeneous backbones, we use a single, deterministic input construction. After preprocessing, each mmWave frame is represented as a feature vector $\mathbf{f}_i[l] \in \mathbb{R}^{256}$. We embed it into $\mathbb{R}^{4096}$ by appending zeros: $\tilde{\mathbf{f}}_i[l] = [\mathbf{f}_i[l]^\top, \mathbf{0}_{3840}^\top]^\top$, i.e., the original 256 features occupy the first 256 entries and the remaining entries are zero-padded. We then reshape $\tilde{\mathbf{f}}_i[l]$ in row-major order into a single-channel $64 \times 64$ map $\tilde{\mathbf{X}}_i[l] \in \mathbb{R}^{1 \times 64 \times 64}$. Equivalently, $\mathbf{f}_i[l]$ fills the top-left $16 \times 16$ block and all other spatial locations are zero.

Stacking $L$ frames yields $\tilde{\mathbf{X}}_i \in \mathbb{R}^{L \times 1 \times 64 \times 64}$. Unless otherwise specified, we fix $L = 4$ for all datasets and models. For 2D CNN backbones (LeNet/VGG/ResNet), we apply the same 2D network to each frame with shared weights and aggregate predictions by temporal averaging of logits: $\mathbf{z}_i = \frac{1}{L} \sum_{l=0}^{L-1} \mathbf{z}_i[l]$ (Palipana et al., 2021). We do not treat $L$ as input channels and do not use 3D convolutions. For recurrent backbones, the per-step input is the original sequence $\{\mathbf{f}_i[l]\}_{l=0}^{L-1}$ (256-D), and for transformer backbones, tokens are extracted from $\tilde{\mathbf{X}}_i[l]$ via patchification.

# E. Baseline Specification and Training Protocol

This appendix documents the baseline model families, naming conventions, a unified input representation, and the training configuration used for all experiments. Our goal is to ensure that all reported differences primarily reflect architectural inductive biases, rather than ad-hoc per-model tuning or inconsistent preprocessing.

## E.1. Baseline families and naming conventions

We evaluate a broad suite of ANN and SNN baselines spanning convolutional, recurrent, and transformer-style architectures that are commonly adopted in wireless sensing pipelines (Yang et al., 2023). Model names follow standard conventions within each family.

**VGG-style CNNs.** We include *VGG9* and *VGG16* following the VGG design principles (Simonyan & Zisserman, 2015). The numeric suffix (e.g., 9 or 16) denotes the total number of *learnable layers* (convolutional + fully connected) in the canonical VGG formulation. Larger suffixes correspond to deeper stacks of convolutional blocks and thus higher representational capacity.

**ResNet-family CNNs.** We use *ResNet18*, *ResNet50*, and *ResNet101*, where the suffix denotes the standard depth of the canonical ResNet variants. Deeper models increase capacity by adding residual blocks while maintaining optimization stability via identity skip connections.

**Transformer-based models.** For transformer-style architectures (e.g., ViT), *depth* is specified by the number of stacked encoder blocks. In this study, we set depth equal to 1 since multi-clocks induce underfitting problem (Yang et al., 2023). Unless otherwise stated, other components (e.g., normalization, MLP ratio, and attention type) follow the default settings of the corresponding baseline implementation.

**Recurrent and hybrid baselines.** Recurrent baselines include RNN, GRU, LSTM, and BiLSTM, as well as hybrid CNN–RNN models (e.g., CNN–GRU) that first extract per-frame features and then aggregate them temporally using a recurrent unit. Unless otherwise stated, we use a single recurrent layer to avoid introducing additional depth-related confounders.

### E.2. Training protocol and settings

**Optimization.** All models are trained for 150 epochs using Adam with initial learning rate $10^{-3}$ and a cosine learning-rate schedule. We use batch size 64 and set weight decay to zero. To reduce randomness, we report results averaged over three random seeds $\{41, 42, 43\}$ using identical data splits and the same training/validation/ evaluation protocol for all models.

**configuration.** All spiking baselines use LIF neurons with firing threshold $v_{\text{th}} = 1.0$, membrane time constant $\beta = 0.5$, reset potential $v_{\text{reset}} = 0$, and input decay enabled. We employ an arctangent surrogate gradient for backpropagation through spikes, and detach the reset operation from the computation graph to improve training stability. Unless explicitly varied in ablations, these neuron and training settings are kept fixed across all SNN baselines to isolate architectural effects. For ANNs and non-spiking temporal models, the temporal axis corresponds to $L$ frames. For SNNs, we simulate LIF dynamics for $T$ discrete timesteps; when a frame-based encoding is used, the input sample is presented over $T$ steps and predictions are obtained by aggregating spike-based outputs over time. Unless stated otherwise, we use $L = 4$ input frames and simulate SNNs with $T = 4$ timesteps in all experiments. In the mechanistic analysis of frequency matching, we use $L = 16$ input frames and simulate SNNs with $T = 16$ timesteps.

**Architectural conventions.** Unless explicitly stated, remaining details (e.g., normalization, activation, residual aggregation, and readout strategy) follow canonical baseline implementations. This design choice avoids introducing configuration artifacts and ensures that observed performance differences are attributable to the model families themselves rather than bespoke hyper-parameter tuning.

## F. Measurement

### F.1. Accuracy

We report top-1 classification accuracy (%) on the test split of each dataset. All results are averaged over three random seeds using the same training, validation, and evaluation protocol for ANN and SNN models. For SNNs, accuracy is computed from spike-based outputs accumulated over $T$ timesteps.

### F.2. Model Size and Parameter Count

Model size is quantified by the total number of learnable parameters. For ANN baselines, this includes all convolutional, fully connected, and normalization layers. For SNNs, we count identical architectural parameters, as spiking neuron dynamics (e.g., membrane potential updates and thresholding) do not introduce additional learnable weights.

### F.3. Operation Count (#OPs)

We report the number of operations per sample (#OPs) as a hardware-agnostic measure of computational complexity. For ANN models, #OPs correspond to the total number of multiply–accumulate (MAC) operations required for a single forward pass. For SNNs, #OPs are computed by counting synaptic accumulate (AC) operations triggered by spikes across all timesteps. Specifically, an AC is counted whenever a presynaptic spike contributes to a postsynaptic membrane potential update. All #OPs are reported in millions (M) and are averaged over the test set. In our setting, the first layer consumes real-valued inputs and is therefore counted as dense MACs, while subsequent spiking layers are counted as spike-driven ACs.

### F.4. Theoretical Energy Estimation

Following the standard operation-level proxy widely adopted for efficiency analysis of ANNs and SNNs (Horowitz, 2014; Yao et al., 2023), we estimate inference-time *compute* energy by weighting operation counts with per-operation energy constants. This proxy is designed for *relative* comparisons and algorithmic insights, rather than absolute chip-level power modeling.

#### F.4.1. ANN ENERGY (MAC-BASED).

For ANNs, each floating-point operation is treated as a multiply-and-accumulate (MAC). Let $\text{FLOPs}_\ell$ denote the number of MACs in layer $\ell$ (equivalently, $\#\text{OP}_\ell$ for ANNs). The theoretical compute energy of layer $\ell$ is

$$\mathcal{E}_\ell^{\text{ANN}} \;=\; E_{\text{MAC}} \cdot \text{FLOPs}_\ell, \tag{99}$$

and the per-sample energy is obtained by summing over layers, $\mathcal{E}^{\text{ANN}} = \sum_\ell \mathcal{E}_\ell^{\text{ANN}}$.

#### F.4.2. SNN ENERGY (SPIKE-DRIVEN AC-BASED).

For SNNs, synaptic computation is predominantly event-driven and consists of spike-triggered *accumulate* (AC) operations. Let $T$ denote the number of simulation timesteps. Under our frame-aligned coding scheme, we set $T = 4$, the resampled temporal length used throughout the paper.

Let $\gamma_\ell(\tau) \in [0, 1]$ denote the dataset-averaged *presynaptic* spike rate entering layer $\ell$. For $\ell \geq 2$, we set $\gamma_\ell(\tau) := r_{\text{spk}}^{(\ell-1)}(\tau)$, where $r_{\text{spk}}^{(\ell-1)}(\tau)$ is the mean fraction of neurons that emit a spike at each timestep in layer $\ell - 1$.

#### F.4.3. FIRST-LAYER CONVENTION AND TOTAL OPERATION COUNT.

Many algorithmic analyses treat all SNN layers as AC-only. In our mmWave setting, however, the input is real-valued, and the first layer is therefore computed densely using MACs. Consistent with our operation counting scheme ($\#\text{OP} = \text{FLOPs}$ for ANN; $\#\text{OP} = \text{FLOPs}_{\ell=1} + \text{SOPs}_{\ell \geq 2}$ for SNN), the total SNN compute energy is

$$\mathcal{E}^{\text{SNN}} \;=\; E_{\text{MAC}} \cdot \text{FLOPs}_1 \;+\; \sum_{\ell=2}^{N_{\text{lyr}}} E_{\text{AC}} \cdot \text{SOPs}_\ell, \tag{100}$$

where $N_{\text{lyr}}$ denotes the number of network layers.

#### F.4.4. OPERATION ENERGY CONSTANTS AND UNIT CONVERSION.

We assume MAC and AC operations are implemented using a $45\,\text{nm}$ CMOS process (Horowitz, 2014; Yao et al., 2023), with

$$E_{\text{MAC}} = 4.6\,\text{pJ}, \qquad E_{\text{AC}} = 0.9\,\text{pJ}. \tag{101}$$

For reporting, note that $1\,\text{J} = 10^{12}\,\text{pJ}$ and $1\,\mu\text{J} = 10^6\,\text{pJ}$, so $\mathcal{E}[\mu\text{J}] = \mathcal{E}[\text{pJ}]/10^6$.

#### F.4.5. LIMITATIONS OF THE PROXY.

This energy estimate is intentionally simplified. It ignores data movement and memory hierarchy effects, control overhead, spike generation and reset costs, as well as hardware-specific implementation details. Moreover, GPU execution does not

faithfully realize event-driven benefits, since binary spikes are typically still represented and processed as floating-point tensors. Substantial latency or throughput gains generally require neuromorphic or spike-optimized hardware backends. Finally, this proxy does *not* rely on any subthreshold or LTI approximation of neuron dynamics; it only uses empirically measured spike rates to estimate spike-triggered synaptic activity.

### F.5. End-to-End Inference Latency

We measure *single-sample* inference latency on real edge devices with batch size 1 under inference mode (no gradient tracking). To ensure a consistent and hardware-aware comparison between ANN and SNN implementations, we measure latency on a *fixed tail subnetwork* that is shared by all models.

#### F.5.1. LATENCY DEFINITION

**Tail-only latency definition.**    Following our deployment implementation, we exclude the network stem and time only the *tail* of the model, defined as the subgraph from the first pooling layer (`pool1`) to the final classifier output (Reddi et al., 2020; Kang et al., 2017). Concretely, the stem—including the initial convolution, normalization, and activation (for ANNs) or the corresponding spiking stem and state reset (for SNNs)—is executed once per run to produce the intermediate activation, but its runtime is not included in the reported latency. The measured latency therefore reflects the computation from `pool1` through all subsequent layers and the final network readout.

For SNNs, the reported latency includes the full spike-based computation within the tail subnetwork, including temporal aggregation across timesteps and the final readout layer, but excludes any task-level output decoding such as softmax or label selection. The same convention is applied to ANN baselines, where latency is measured up to the pre-softmax logits.

For a given input sample $\mathbf{X}_i$, we denote the measured tail latency as

$$t_{\text{tail}}(\mathbf{X}_i). \tag{102}$$

**Rationale for tail-only measurement.**    We adopt a tail-only latency definition for two practical reasons. First, on neuromorphic hardware such as Darwin3 (Ma et al., 2024), the initial layers operate on real-valued inputs and are executed on the host processor for input adaptation and spike encoding, as floating-point convolutions and normalizations are not natively supported on-chip. This input interface is therefore system- and implementation-dependent, and does not reflect the intrinsic efficiency of spike-based inference on the neuromorphic device. Second, excluding the stem removes device-specific front-end effects and enables a fair and hardware-aware comparison by measuring only the shared subnetwork that can be efficiently executed across ANN and SNN platforms.

#### F.5.2. NETWORK TIME FOR ANN AND SNN

For ANN models, the tail latency corresponds to a single forward pass through the tail subnetwork:

$$t_{\text{tail}}^{\text{ANN}}(\mathbf{X}_i) \;=\; t_{\text{forward}}(\mathbf{X}_i). \tag{103}$$

For SNN models simulated over $T$ discrete timesteps, the tail latency includes the full spike-based temporal simulation and readout over all timesteps, and can be expressed as

$$t_{\text{tail}}^{\text{SNN}}(\mathbf{X}_i; T) \;=\; \sum_{t=1}^{T} t_{\text{step}}(\mathbf{X}_i, t), \tag{104}$$

where $t_{\text{step}}(\mathbf{X}_i, t)$ denotes the per-timestep computation time within the tail, including synaptic accumulation, neuron state updates, and readout computation. This formulation explicitly captures the temporal nature of SNN inference and ensures that ANN and SNN latencies are compared under an identical tail-level protocol.

#### F.5.3. TIMING PROTOCOL

To obtain reliable wall-clock measurements, we adopt the following protocol. First, we perform several warm-up runs to amortize one-time overheads such as kernel loading, memory allocation, and runtime initialization; unless otherwise stated, warm-up is enabled. Second, for devices with asynchronous execution (e.g., GPU-accelerated platforms), explicit

synchronization is enforced immediately before and after the timed region to prevent underestimation of execution time. After warm-up, latency is recorded over $M$ repeated inference runs and summarized by the empirical average

$$\bar{t}_{\text{tail}} \triangleq \frac{1}{M} \sum_{m=1}^{M} t_{\text{tail}}^{(m)}, \tag{105}$$

where $t_{\text{tail}}^{(m)}$ denotes the $m$-th measured tail latency.

All latency measurements use identical implementation settings for ANN and SNN baselines on the same device, including batch size, threading configuration, and numerical precision. Detailed device specifications and measurement configurations are reported in Appendix J.

## G. Supplementary Validation Experiments on Accuracy

To assess the generality of our findings, we further evaluate SNNs with advanced architectures, including *SEW-ResNet* (Fang et al., 2021a), *MS-ResNet* (Hu et al., 2024), *Spikformer* (Zhou et al., 2023), *SDT* (Yao et al., 2024a), *QKFormer* (Zhou et al., 2024), and *SpikingResFormer* (Shi et al., 2024). Table 4 summarizes the accuracy–capacity trade-off of ANN and SNN baselines. Across all four datasets, SNN backbones consistently surpass ANNs at comparable parameter budgets, suggesting that spike-based temporal dynamics provide a more suitable inductive bias for mmWave sequences than purely frame-based processing. Under the same unified preprocessing and training/evaluation protocol (mean over three seeds), SpikingLeNet already improves over the per-dataset best ANN by **6.22%** relative accuracy on average, and the strongest SNN improves over the strongest ANN by **17.50%** on average, without increasing capacity beyond the reported parameter counts. We attribute this advantage to the inherent low-pass temporal filtering and dynamic integration of LIF neurons, which better aligns with the discriminative spectral structure of mmWave signals. MMActivity is a notable exception: its smaller sample size reduces the benefit of larger models, making simpler backbones comparatively strong. Finally, spiking attention models offer an attractive accuracy–compactness operating point, e.g., SpikFormer attains $91.29\%$ on AOPHand with 2.57M parameters, and QKFormer achieves comparable performance with only 2.41M parameters, indicating that lightweight spiking transformers can be markedly more parameter-efficient than large CNNs.

**Recurrent baselines: capacity and fairness.**   To ensure a fair comparison across heterogeneous backbones, we adopt a *unified* recurrent configuration and training protocol across *all* datasets and do not retune the RNN/GRU/LSTM architectures on a per-dataset basis. Each recurrent model consumes the same per-frame input representation (a $64\times64$ map, flattened to a vector per timestep) and uses a lightweight recurrent core followed by a linear classifier. While this unified setting yields relatively low accuracy on mmFiT (Table 1), the recurrent baselines are not trivially under-powered: under the *same* configuration, GRU achieves the best ANN accuracy on Pantomime, LSTM reaches $72.47\%$ on Pantomime, and a vanilla RNN attains $48.33\%$ on MMActivity. These cross-dataset results indicate that the recurrent implementations are functional and competitive under our shared protocol, and that the mmFiT outcome is more consistent with a representation–model mismatch (or dataset-specific temporal dynamics) than with an intentionally under-parameterized baseline.

**Critical-Difference Diagram.**   Figure 8 reports the critical-difference (CD) diagram over 12 evaluation blocks (4 datasets $\times$ 3 seeds), where methods are ordered by average rank (lower is better) under the Friedman–Nemenyi protocol. A clear separation emerges between spiking and non-spiking baselines: top positions are consistently occupied by LIF-based SNNs. In particular, **SpikFormer** achieves the best overall rank ($\bar{r} \approx 3.54$), followed by **SpikingResFormer** ($\bar{r} \approx 4.00$) and **MS-ResNet50** ($\bar{r} \approx 4.04$); these models form the leading clique in CD plot, indicating statistically indistinguishable performance at the chosen significance level. Importantly, even the lightweight **SpikingLeNet** (mean accuracy $\sim 77.7\%$, $\bar{r} \approx 8.92$) consistently outranks the strongest ANN baselines (e.g., VGG/ResNet families with $\bar{r} \approx 12\text{–}14$), demonstrating that the benefit is not solely attributable to architectural scaling. In contrast, several ANN sequence models (e.g., RNN/LSTM/BiLSTM) exhibit substantially worse average ranks, reflecting poorer robustness across datasets and seeds. Overall, these supplementary results corroborate that mmWave recognition benefits from the spiking temporal processing bias induced by LIF neurons, rather than capacity alone, and motivate our subsequent analyses on efficiency (energy/#OPs) and on how frequency-domain alignment explains the observed accuracy gains of SNNs over ANNs.

*Table 4.* Overall accuracy (%) and parameter count (M) of ANN and SNN baselines across datasets. Results are reported as mean values over *three* runs with different random seeds. **Bold** / **Bold** denote the best accuracy, and Underline / Underline denote the second-best accuracy, for ANN-based and SNN-based baselines, respectively, on each dataset.

| Baseline | AOPHand Accuracy (%) | AOPHand Params (M) | mmFiT Accuracy (%) | mmFiT Params (M) | Pantomime Accuracy (%) | Pantomime Params (M) | MMActivity Accuracy (%) | MMActivity Params (M) |
|---|---|---|---|---|---|---|---|---|
| MLP | $69.70_{\pm0.99}$ | 4.327 | $66.51_{\pm2.02}$ | 4.327 | $66.81_{\pm0.57}$ | 4.328 | $60.83_{\pm2.89}$ | 4.327 |
| LeNet | $60.86_{\pm1.68}$ | 4.191 | $62.36_{\pm0.58}$ | 4.192 | $61.83_{\pm0.42}$ | 4.196 | $59.17_{\pm3.82}$ | 4.191 |
| VGG9 | $74.39_{\pm0.61}$ | 31.603 | $69.36_{\pm2.15}$ | 31.605 | $72.63_{\pm0.46}$ | 31.622 | $70.00_{\pm4.33}$ | 31.601 |
| VGG16 | $67.92_{\pm6.02}$ | 70.313 | $65.43_{\pm0.58}$ | 70.315 | $71.60_{\pm1.88}$ | 70.331 | $62.50_{\pm4.33}$ | 70.311 |
| ResNet18 | $72.47_{\pm1.30}$ | 11.173 | $71.94_{\pm1.29}$ | 11.174 | $72.63_{\pm0.59}$ | 11.178 | $61.67_{\pm6.29}$ | 11.173 |
| ResNet50 | $72.54_{\pm0.80}$ | 23.514 | $71.84_{\pm1.97}$ | 23.516 | $73.90_{\pm0.43}$ | 23.532 | $61.67_{\pm2.89}$ | 23.512 |
| ResNet101 | $69.70_{\pm0.72}$ | 42.506 | $72.64_{\pm0.25}$ | 42.508 | $73.85_{\pm0.29}$ | 42.525 | $64.17_{\pm1.44}$ | 42.504 |
| RNN | $26.40_{\pm9.25}$ | **0.026** | $22.56_{\pm5.93}$ | **0.026** | $20.84_{\pm0.88}$ | **0.027** | $48.33_{\pm1.44}$ | **0.025** |
| GRU | $67.52_{\pm5.73}$ | 0.075 | $14.11_{\pm0.38}$ | 0.075 | $75.45_{\pm0.99}$ | 0.076 | $47.50_{\pm2.50}$ | 0.075 |
| LSTM | $20.71_{\pm1.12}$ | 0.100 | $13.14_{\pm0.52}$ | 0.100 | $72.47_{\pm3.29}$ | 0.101 | $54.17_{\pm7.64}$ | 0.100 |
| BiLSTM | $46.14_{\pm1.40}$ | 0.200 | $12.98_{\pm0.89}$ | 0.200 | $73.67_{\pm1.39}$ | 0.203 | $45.83_{\pm3.82}$ | 0.200 |
| CNN-GRU | $61.98_{\pm9.44}$ | 0.463 | $67.80_{\pm0.49}$ | 0.463 | $72.77_{\pm2.36}$ | 0.464 | $65.00_{\pm2.50}$ | 0.463 |
| ViT | $21.39_{\pm3.31}$ | 2.176 | $36.40_{\pm5.65}$ | 2.176 | $42.16_{\pm6.07}$ | 2.178 | $65.83_{\pm5.77}$ | 2.175 |
| SpikingLeNet | $83.70_{\pm4.24}$ | 4.191 | $73.67_{\pm1.55}$ | 4.192 | $78.31_{\pm0.50}$ | 4.196 | $75.00_{\pm6.61}$ | 4.191 |
| SpikingVGG9 | $88.51_{\pm0.40}$ | 31.603 | $75.93_{\pm1.82}$ | 31.605 | $82.77_{\pm1.06}$ | 31.622 | $75.83_{\pm6.29}$ | 31.601 |
| SpikingVGG16 | $86.86_{\pm0.60}$ | 70.313 | $70.65_{\pm2.51}$ | 70.315 | $82.82_{\pm0.99}$ | 70.331 | $65.83_{\pm2.89}$ | 70.311 |
| SEW-ResNet18 | $88.97_{\pm0.64}$ | 11.173 | $78.57_{\pm1.21}$ | 11.174 | $84.93_{\pm0.37}$ | 11.178 | $66.67_{\pm3.82}$ | 11.173 |
| SEW-ResNet50 | $86.01_{\pm1.02}$ | 23.514 | $76.20_{\pm1.17}$ | 23.516 | $85.02_{\pm1.10}$ | 23.532 | $65.83_{\pm1.44}$ | 23.512 |
| MS-ResNet18 | $90.96_{\pm0.31}$ | 11.178 | $85.08_{\pm0.73}$ | 11.178 | $88.50_{\pm0.53}$ | 11.182 | $65.83_{\pm3.82}$ | 11.177 |
| MS-ResNet50 | $89.51_{\pm0.69}$ | 23.514 | $85.41_{\pm0.83}$ | 23.516 | $89.11_{\pm0.64}$ | 23.532 | $67.50_{\pm4.33}$ | 23.512 |
| SpikFormer | $91.29_{\pm1.58}$ | 2.566 | $83.58_{\pm0.52}$ | 2.567 | $87.19_{\pm0.99}$ | 2.569 | $75.83_{\pm1.44}$ | 2.566 |
| SDT | $88.31_{\pm0.52}$ | 6.081 | $82.87_{\pm1.32}$ | 6.081 | $86.10_{\pm0.99}$ | 6.084 | $69.17_{\pm3.82}$ | 6.080 |
| QKFormer | $90.17_{\pm0.58}$ | 2.414 | $83.36_{\pm0.70}$ | 2.414 | $87.14_{\pm0.82}$ | 2.416 | $66.67_{\pm2.89}$ | 2.414 |
| SpikingResFormer | $92.74_{\pm0.64}$ | 17.310 | $86.38_{\pm0.74}$ | 17.310 | $88.45_{\pm0.74}$ | 17.315 | $65.83_{\pm3.82}$ | 17.309 |

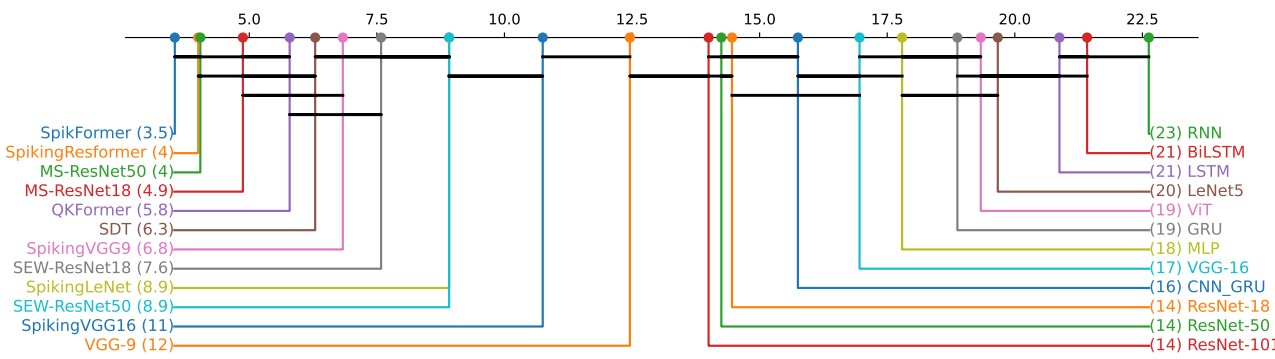

*Figure 8.* Critical-difference (CD) diagram for ANN baselines vs. SNN baselines. Average ranks are computed over dataset×seed blocks (lower is better); thick horizontal bars denote groups that are not significantly different under a Nemenyi post-hoc test at $\alpha = 0.05$.

# H. Supplementary Energy-Efficiency Experiments

Table 5 summarizes dataset-averaged inference energy and operation counts under a unified measurement protocol, revealing two consistent trends. First, ANN baselines exhibit a largely capacity-driven scaling where energy closely tracks #OPs: lightweight temporal models (e.g., RNN/GRU/LSTM) remain the most compute-frugal, whereas deep CNNs (VGG/ResNet-101) incur substantially higher #OPs and energy; moreover, for a fixed ANN architecture, the reported values are nearly invariant across datasets, as the forward graph and tensor shapes are fixed once the input specification is set. Second, SNN+LIF models deliver pronounced efficiency gains relative to comparable architectural families: for instance, SpikingLeNet operates at $1.45$–$2.53\,\mu$J with $\approx 0.94$–$1.43$ M #OPs, versus $251.08\,\mu$J and $54.58$ M for LeNet (about $99\times$ lower energy and $\sim 52\times$ fewer operations), and it is also lower than the most efficient ANN baseline (RNN at $7.35\,\mu$J and $1.60$ M). Within the SNN group, efficiency spans a clear frontier: compact spiking CNNs are the most energy-minimal, while modern spiking backbones (e.g., SEW-ResNet18 and QKFormer) occupy a low-to-mid regime ($\approx 28$–$42\,\mu$J with $\approx 31$–$36$ M), and heavier spiking CNNs/transformer-style models are costlier. Finally, the mild cross-dataset variation observed for SNNs

*Table 5.* Energy ($\mu$J) and #OPs (M) of ANN baselines and SNN+LIF models aggregated over datasets (averaged over three seeds). **Bold** denotes the lowest (best), and Underline denotes the second-lowest, within ANN-based and SNN-based baselines, respectively, on each dataset (see Appendix F).

| Baseline | AOPHand | | mmFiT | | Pantomime | | MMActivity | |
|---|---|---|---|---|---|---|---|---|
| | Energy | #OPs | Energy | #OPs | Energy | #OPs | Energy | #OPs |
| MLP | 19.90 | 4.33 | 19.90 | 4.33 | 19.91 | 4.33 | 19.90 | 4.33 |
| LeNet | 251.08 | 54.58 | 251.08 | 54.58 | 251.10 | 54.59 | 251.08 | 54.58 |
| VGG9 | 3352.70 | 728.85 | 3352.71 | 728.85 | 3352.79 | 728.87 | 3352.69 | 728.85 |
| VGG16 | 6017.25 | 1308.10 | 6017.26 | 1308.10 | 6017.34 | 1308.12 | 6017.24 | 1308.10 |
| ResNet18 | 655.24 | 142.44 | 655.22 | 142.44 | 655.24 | 142.44 | 655.22 | 142.44 |
| ResNet50 | 1522.02 | 330.87 | 1522.02 | 330.87 | 1522.10 | 330.89 | 1522.01 | 330.87 |
| ResNet101 | 2923.68 | 635.58 | 2923.69 | 635.58 | 2923.76 | 635.60 | 2923.67 | 635.58 |
| RNN | 7.35 | 1.60 | 7.35 | 1.60 | 7.36 | 1.60 | 7.35 | 1.60 |
| GRU | 22.20 | 4.83 | 22.20 | 4.83 | 22.20 | 4.83 | 22.20 | 4.83 |
| LSTM | 29.55 | 6.42 | 29.55 | 6.42 | 29.55 | 6.42 | 29.54 | 6.42 |
| BiLSTM | 59.09 | 12.85 | 59.10 | 12.85 | 59.10 | 12.85 | 59.10 | 12.85 |
| CNN-GRU | 128.55 | 27.95 | 128.55 | 27.95 | 128.56 | 27.95 | 128.55 | 27.95 |
| ViT | 82.61 | 17.96 | 82.61 | 17.96 | 82.62 | 17.96 | 82.61 | 17.96 |
| SpikingLeNet | 2.53 | 1.04 | 2.04 | 0.94 | 2.44 | 0.94 | 1.45 | 1.43 |
| SpikingVGG9 | 266.47 | 293.87 | 155.13 | 170.71 | 155.79 | 170.86 | 302.31 | 335.52 |
| SpikingVGG16 | 2044.59 | 2269.56 | 1723.84 | 1913.74 | 1825.96 | 2026.68 | 1348.99 | 1498.50 |
| SEW-ResNet18 | 41.89 | 35.67 | 39.00 | 34.74 | 37.56 | 30.85 | 33.46 | 35.54 |
| SEW-ResNet50 | 127.72 | 131.02 | 141.30 | 136.16 | 119.79 | 122.16 | 105.72 | 115.80 |
| MS-ResNet18 | 66.93 | 20.99 | 63.48 | 19.21 | 65.38 | 19.27 | 49.57 | 17.51 |
| MS-ResNet50 | 562.09 | 137.47 | 561.30 | 148.41 | 561.70 | 137.14 | 467.72 | 115.80 |
| SpikFormer | 181.12 | 200.14 | 174.28 | 192.82 | 188.59 | 208.43 | 174.09 | 193.24 |
| SDT | 174.60 | 155.74 | 168.11 | 151.65 | 160.16 | 141.06 | 169.04 | 152.54 |
| QKFormer | 33.17 | 35.76 | 30.57 | 33.14 | 33.06 | 35.63 | 28.77 | 31.77 |
| SpikingResFormer | 435.84 | 473.20 | 372.92 | 405.72 | 473.70 | 515.34 | 452.20 | 500.77 |

(e.g., SpikingLeNet ranging from $1.45$ to $2.53\,\mu$J) suggests that data-dependent spike activity affects the realized energy, highlighting that SNN efficiency is jointly governed by architecture and activation sparsity rather than by compute alone.

## I. Implementation of Low-Pass Filter for LeNet

To isolate the effect of a purely input-agnostic spatial smoothing baseline, we implement an explicit low-pass filter that operates on each per-frame sensing map $\mathbf{X}_i[l] \in \mathbb{R}^{C \times H \times W}$ before feeding the sequence $\mathbf{X}_i \in \mathbb{R}^{L \times C \times H \times W}$ into an ANN backbone (e.g., LeNet). The filter is applied independently to each frame and each channel, i.e., it does not couple different $l$'s and does not use any information from the class label $y_i$.

**Scope of the ablation.** The explicit low-pass baseline considered in this work is deliberately restricted to a *static, input-agnostic spatial* filter applied independently to each frame $\mathbf{X}_i[l] \in \mathbb{R}^{C \times H \times W}$. Its sole purpose is to test whether uniform attenuation of high spatial-frequency components within individual frames can explain the observed performance gains. We intentionally avoid introducing an explicit temporal low-pass filter along the frame index $l \in \{0, \ldots, L-1\}$. This choice reflects the nature of the available mmWave datasets: each frame $\mathbf{X}_i[l]$ is not a raw continuous-time radar return, but the output of a fixed sensing and preprocessing pipeline (e.g., frame accumulation, windowing, and clutter suppression) (Tiwari, 2023). Consequently, the temporal axis already represents a rate-limited, discretized observation sequence with task-dependent semantics. Applying an additional temporal low-pass filter at this level would introduce a new, hand-crafted temporal inductive bias and an extra time constant that is not intrinsic to the original data, thereby altering the task semantics and confounding the comparison. We further restrict the spatial filter to the simplest hard radial mask. More sophisticated designs, such as higher-order filters, learned frequency responses, or adaptive spatial kernels, would introduce extra parameters, computational overhead, and dataset-specific tuning, effectively embedding additional inductive bias or learning capacity into the ANN baseline rather than isolating the effect of low-pass smoothing tself (Dwivedi et al., 2023; Wu et al., 2023; Jia et al., 2016; Proakis & Manolakis, 2007).

In contrast, SNNs with LIF neurons inherently implement a stateful temporal low-pass mechanism through their membrane dynamics, controlled by the decay factor $\beta \in (0, 1)$. This mechanism enables temporal integration and event-driven sparsity without introducing extra trainable parameters or preprocessing cost. The present ablation therefore serves as a conservative validity check: it demonstrates that static spatial smoothing alone is insufficient to replicate the ehavior induced by LIF temporal dynamics, especially when task-relevant structure is distributed across the discriminative frequency band $\Omega^\star$.

**Radial frequency mask.** For a given spatial resolution $(H, W)$, we construct a fixed *radial* hard low-pass mask in the 2D spatial-frequency domain. Concretely, we form normalized discrete spatial-frequency coordinates $(\xi, \eta) \in [-0.5, 0.5]^2$, where $[-0.5, 0.5]^2 \triangleq [-0.5, 0.5] \times [-0.5, 0.5]$ denotes the Cartesian product of the normalized frequency axes along the width and height dimensions, respectively. The coordinates are obtained using fftfreq followed by fftshift, such that the zero-frequency (DC) component is centered and $0.5$ corresponds to the Nyquist limit (half the sampling rate) under this normalization (Hsu, 2011).

We then define the radial spatial-frequency magnitude as

$$\rho(\xi, \eta) \triangleq \sqrt{\xi^2 + \eta^2}, \tag{106}$$

and choose a cutoff radius $\nu_{\mathrm{lp}} \in [0, 0.5]$, expressed as a fraction of the Nyquist limit. The resulting hard low-pass mask is

$$M_{\mathrm{lp}}(\xi, \eta) \triangleq \Theta\big[\nu_{\mathrm{lp}} - \rho(\xi, \eta)\big], \tag{107}$$

so that spatial-frequency components with $\rho(\xi, \eta) \leq \nu_{\mathrm{lp}}$ are retained, while higher-frequency components are suppressed.

**FFT-domain filtering.** For each channel map $x \in \mathbb{R}^{H \times W}$ (i.e., one slice of $\mathbf{X}_i[l]$), we apply the filter in the spatial-frequency domain via the standard two-dimensional discrete Fourier transform (2D DFT). Here, a 2D DFT is obtained by applying the one-dimensional discrete Fourier transform separately along the two spatial dimensions $(H, W)$; the plural form DFTs refers only to multiple such transform operations and does not introduce a distinct operator. In practice, fft2 and ifft2 denote the forward and inverse 2D DFTs, while fftshift and ifftshift rearrange the frequency components to and from a centered (DC-aligned) representation (Proakis & Manolakis, 2007). The filtered output is given by

$$\tilde{x} = \Re\Big\{ \mathrm{ifft2}\big( \mathrm{ifftshift}\big( \mathrm{fftshift}(\mathrm{fft2}(x)) \odot M_{\mathrm{lp}} \big) \big) \Big\}, \tag{108}$$

where $\odot$ denotes elementwise multiplication and $\Re\{\cdot\}$ takes the real part after the inverse transform. Once constructed for a fixed $(H, W)$ and $\nu_{\mathrm{lp}}$, the same mask $M_{\mathrm{lp}}$ is reused for all samples $i$, all frames $l \in \{0, \ldots, L-1\}$, and all channels.

## J. Deployment Devices

To assess real-world deployability, we benchmark inference latency across a diverse set of edge platforms, including GPU-accelerated embedded SoCs, a CPU-only single-board computer, and a neuromorphic processor. This selection covers representative deployment scenarios ranging from general-purpose edge AI acceleration to event-driven neuromorphic inference.

Table 6 summarizes the exact hardware configurations used in our measurements, including the module SKU and configured `nvpmodel` power mode for NVIDIA Jetson devices (NVI, 2024), the Raspberry Pi (Ras, 2019) generation and RAM capacity, and the Darwin3 board and runtime revision. Figure 9 shows photographs of the evaluated platforms, providing a visual reference for the physical deployment setups and the distinct hardware execution models, particularly the on-chip spike-based processing on the Darwin3 neuromorphic board. All latency benchmarks are implemented using Python-based test scripts that are executed directly on each target device, with wall-clock time recorded on-device. Unless otherwise stated, all latency measurements are conducted with batch size 1, identical numerical precision, and are averaged over multiple runs. We fix $m = 1000$, and the number of warm-up runs is set to 50 for all latency experiments, and additionally report percentile statistics to characterize runtime variability.

**Embedded GPU and CPU platforms.** The NVIDIA Jetson Orin family (Nano, NX, and AGX) represents modern GPU-accelerated edge SoCs widely used for low-latency and power-constrained inference. For these devices, we fix the power mode using `nvpmodel` and report the exact configuration in Table 6 to ensure reproducibility. Raspberry Pi 4 is included as a CPU-only baseline platform; following standard deployment practice, it is treated as a general-purpose processor without any dedicated neural network accelerator.

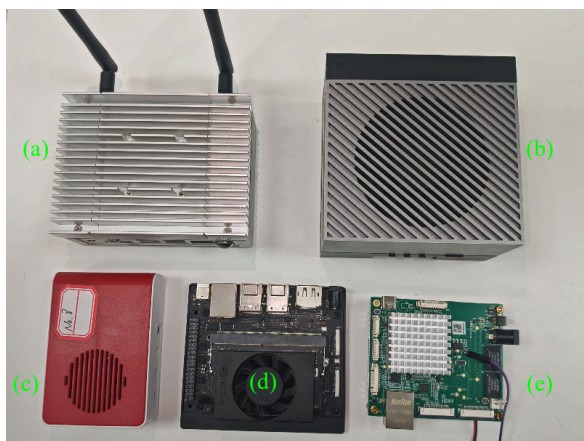

*Figure 9.* Physical deployment devices used for end-to-end latency evaluation: (a) NVIDIA Jetson Orin NX, (b) NVIDIA Jetson AGX Orin, (c) Raspberry Pi 4 (CPU-only), (d) NVIDIA Jetson Orin Nano, (e) Darwin3 neuromorphic board. All devices are evaluated under the runtime configurations reported in Table 6.

**Neuromorphic deployment on Darwin3.** Darwin3 is a neuromorphic processor designed for event-driven spike-based computation. On this platform, only spike-based operations are executed on-chip, including synaptic accumulation, neuron state updates, and spike-based readout. Layers operating on real-valued inputs (e.g., the initial convolutional stem) are executed on the host processor to perform input adaptation and spike encoding, as floating-point operations are not natively supported by the neuromorphic cores. Accordingly, latency measurements on Darwin3 include only the on-chip execution time of the spike-based tail subnetwork, consistent with the tail-only latency definition adopted in Appendix F.5.

*Table 6.* Deployment devices used for latency benchmarking. We report the exact SKU and configured power mode to ensure reproducibility.

| Device (exact SKU) | CPU | GPU / NPU | Memory / Power mode |
|---|---|---|---|
| Raspberry Pi 4 Model B (8GB) | 4× Cortex-A72 | CPU-only (no NN accelerator) | 8GB LPDDR4 / – |
| Jetson Orin NX 16GB | 8× Cortex-A78AE | Ampere (1024 CUDA / 32 Tensor) | 16GB LPDDR5 / 25W |
| Jetson AGX Orin 32GB | 8× Cortex-A78AE | Ampere (1792 CUDA / 56 Tensor) | 32GB LPDDR5 / 30W |
| Jetson Orin Nano Super Dev Kit | 6× Cortex-A78AE | Ampere (1024 CUDA / 32 Tensor) | 8GB LPDDR5 / 25W |
| Darwin3 board (rev. X) | – | Neuromorphic cores | on-chip memory |

## K. Discussion

**More Complicated mmWave Scenarios.** Beyond controlled laboratory settings, practical mmWave sensing systems operate in environments that are both physically complex and semantically heterogeneous. Real-world signals are shaped by rich multipath propagation, dynamic occlusions, platform motion, and increasingly, by actively reconfigurable or adversarial elements. For example, recent studies show that intelligent reflecting surfaces (IRS) (Guo et al., 2025), while beneficial for non-line-of-sight enhancement, can be intentionally manipulated to induce non-stationary, frequency-selective distortions in the received temporal signal, leading to severe sensing failures. Such effects cannot be modeled as simple additive noise, but rather act as implicit, time-varying spectral reshaping operators. A complementary source of complexity arises from semantic interference and cross-modal inconsistency. In autonomous driving and urban scenarios (Wang et al., 2024), visually realistic but physically non-living objects (e.g., billboards or projections) can strongly confound perception pipelines. Recent work demonstrates that fusing mmWave radar with vision can mitigate such failures by exploiting modality-specific physical signatures (e.g., radar cross section) that remain stable under visual interference. While effective, these end-to-end fusion approaches rely on task- and architecture-specific feature learning, and provide limited interpretability into why certain temporal or spectral components are informative. From the perspective of our framework, both adversarial propagation effects and semantic interference can be unified as transformations of the effective discriminative spectrum observed by the model. Our dataset-level diagnostics and frequency-matching analysis abstract away spatial layouts and sensing modalities, and instead characterize how discriminative mass is distributed over temporal frequencies. This abstraction

remains meaningful under complex environments, but also reveals its limitations: highly dynamic scenes may induce multi-band, time-varying spectra for which a single fixed low-pass operating point is suboptimal. These observations motivate future extensions toward adaptive or robust frequency-matching mechanisms—potentially informed by multi-modal cues—while preserving the analysis-driven design principles and energy-efficiency advantages of spiking dynamics.

**Method Generality beyond mmWave.** Although our empirical analysis is instantiated on mmWave sensing, the proposed frequency-aware SNN analysis is not modality-specific. It applies to any sensing pipeline where (i) the input spectrum contains non-trivial high-frequency noise, while (ii) discriminative cues also reside in the high-frequency band. Under this regime, our analysis provides a principled way to characterize and control the trade-off between noise suppression and information preservation, rather than relying on ad-hoc low-pass heuristics. This suggests immediate applicability to other interference-prone modalities (e.g., Radio Frequency (RF) (Yang et al., 2022), acoustic (Lapins et al., 2024), vibration (Pan et al., 2025), Wi-Fi Channel State Information (CSI) (Miao et al., 2025) and event-like streams (Johnston et al., 2025)), where similar spectral entanglement between noise and task-relevant content is common.

**On the Choice of LIF Variants** Beyond the standard LIF neuron considered in the main paper, we conducted a preliminary study of several LIF variants within the same LeNet backbone and evaluated them across all four mmWave datasets. Figure 10 summarizes their classification accuracy. We observe that certain variants, such as PLIF (Fang et al., 2021b), can achieve competitive or even improved performance on specific datasets. However, their behavior is notably less consistent across datasets than vanilla LIF, with accuracy gains on some datasets accompanied by degradation on others. These results suggest that while more flexible neuron dynamics may increase representational capacity, they also introduce stronger dataset dependence. In contrast, the standard LIF exhibits more stable performance, consistent with our analysis that emphasizes principled data–model spectral matching rather than increased neuronal complexity. A comprehensive investigation of alternative spiking neuron models is beyond the scope of this work. Nevertheless, this preliminary evidence highlights an important direction for future research: systematically characterizing the temporal and spectral properties of different LIF variants (e.g., PLIF, GLIF (Yao et al., 2022), PSN (Fang et al., 2023)), and analyzing how their intrinsic dynamics align with dataset-specific discriminative frequency structures. Such an analysis could further generalize the data–model matching framework developed in this paper and guide informed neuron-model selection for mmWave sensing and other similar datasets.

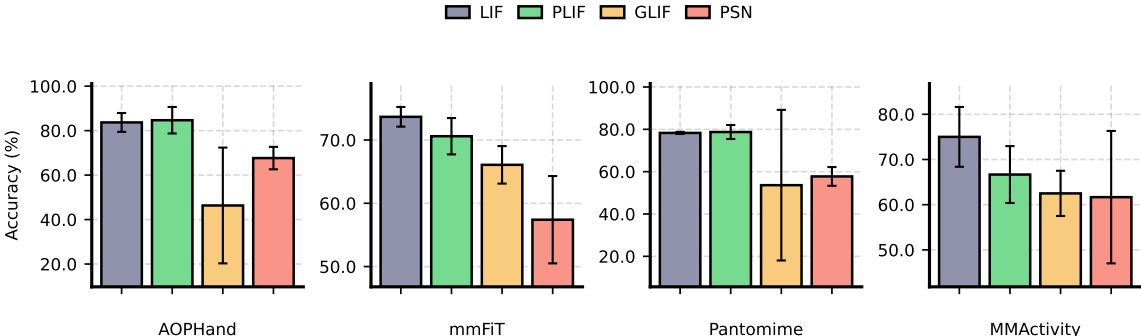

*Figure 10.* Classification accuracy of LeNet-based SNNs equipped with different spiking neuron variants across four mmWave datasets. While certain variants (e.g., PLIF) achieve strong performance on specific datasets, their accuracy varies substantially across datasets. In contrast, the standard LIF neuron exhibits more consistent performance, supporting the importance of stable data–model spectral matching over increased neuronal complexity.

**From model-level evaluation to end-to-end deployment.** Our current system study focuses on model inference and has been validated on five edge devices with consistent latency measurements. A natural next step is to extend the evaluation boundary to the full sensing-to-decision stack (Jin et al., 2024; Zhao et al., 2023), where practical factors such as mmWave antenna array configuration, signal conditioning, channel decoding, and on-device pre-processing can shift the effective spectrum presented to the network. In this setting, the proposed analysis can serve as an interface between signal processing choices and SNN dynamics, enabling joint optimization across the front-end and the inference back-end.

**Platform outlook and neuromorphic deployment.** Beyond general-purpose edge hardware, the proposed methodology is compatible with emerging neuromorphic and specialized platforms. In particular, we have deployed our models on the Darwin3 chip and observed inference latency comparable to ANN deployments on commodity devices, indicating that the gap between SNN deployment and practical latency constraints can be significantly reduced with appropriate software–hardware co-design (Yao et al., 2024b). This opens up a broader design space for future work: co-optimizing representation, SNN parameters, and runtime scheduling for memory/latency/energy targets on constrained accelerators.

