# OpenReview forum: "Frequency Matching in Spiking Neural Networks for mmWave Sensing"
_ICML.cc/2026/Conference — ICML 2026 regular_

### Official Review · Reviewer_Smez · 2026-02-22

**Soundness:** 2
**Presentation:** 4
**Significance:** 3
**Originality:** 3
**Overall Recommendation:** 5
**Confidence:** 3

**Summary:**

This paper studies utilizing SNNs for energy-efficient sensing utilizing mmWave signals. Since the leaky integerate-and-fire mechanism is an inherent lowpass filter, this works to mitigate the high-frequency noise that is prevalant for mmWave signals. The authors propose an algorithm that decides the membrane decay factor based on the data's spectral content. Results show that the proposed method performs better than conventional NN-based methods in terms of both accuracy and energy efficiency, and furthermore demonstrates that by utilizing appropriate hardware (Neuromorphic chips), the latency could be comparable to that of conventional NNs.

**Compliance With Llm Reviewing Policy:**

Affirmed.

**Final Justification:**

The author's third rebuttal has cleared the most obvious obstacle to implement this in real world settings. Thus, I am increasing my score.

**Key Questions For Authors:**

Q1: For the previous ANN baselines utilizing separate LPFs, the authors note that the most significant source of accuracy drop comes from using a hard LPF. However, wouldn't the comparison be flawed in this case? The reviewer thinks that a matched FIR filter that somehow mimics the freqeuncy response of the overall SNN architecture would be the best comparison for this pre-processed ANN baseline. Moreover, if the filter characteristics of the preprocessing module for the ANN is left to be learned by the neural network, we may see a larger performance improvement.

Q2: While SNNs can enable extremely low power consumption, the reviewer understands that the membrane potentials should be saved for the T = 4 (or other) timesteps, which would require significant memory. Since this work is oriented for edge devices, wouldn't this be a limiting factor? The reviewer is curious if the memory utilization is within acceptable limits.

Q3: The authors proposed an algorithm that sets a upper bound of $\beta$, in which further increase in the LPF strength would diminish performance. While the derivation of the upper bound is useful, the reviewer thinks that it is more important to propose an algorithm that at least recommends a rougn version of $\beta$. Do the authors have a way to exactly recommend a specific value of $\beta$?

**Limitations:**

Yes

**Strengths And Weaknesses:**

Soundness: The evaluations and experiments are mostly sound, but the reviewer has found some flaws for the previous ANN comparisons.

Presentation: The paper is overall well-polished, with equations easy to understand. Some improvements to the figure visibility could help in better polishing this manuscript.

Significance: The significance is rather high, with a possibility of being used for edge mmWave radar devices.

Originality: The membrane decay factor algorithm is original, and the overall results confirm that the algorithm works as intended.

---

> ### Author Rebuttal · Authors · 2026-03-31
>
> We thank the reviewer for the positive assessment of our originality and clarity. Our responses to the specific queries follow.
>
> > **Q1. Fairness of the Comparison.**
>
> We agree that a matched FIR filter or a learnable ANN front-end could serve as a stronger ANN-side baseline and may further improve ANN performance. However, the purpose of our ANN+LPF experiment is not to exhaust ANN front-end designs or to provide an oracle upper bound, but to perform a controlled diagnostic test: whether external low-pass smoothing alone can reproduce the benefit of intrinsic LIF dynamics.
>
> Our results suggest that it cannot fully do so. As shown in Fig. 5 of the paper, fixed LPF preprocessing improves LeNet at most from 59.17% to 66.67% on MMActivity. Following the reviewer’s suggestion, we additionally evaluated an ANN baseline with learnable front-end filters based on PLFNets [1], a radar/RF-oriented parameterized filtering module. Over three random seeds, this further improved accuracy to 66.39% and 72.50%, respectively. However, these ANN variants still remained below SpikingLeNet.
>
> | Dataset    | LeNet | LeNet+LPF | LeNet+PLFNets | SpikingLeNet |
> | ------------ | ------: | ----------: | --------------: | -------------: |
> | Pantomime  |      $61.83_{\pm0.42}$ |          $65.49_{\pm2.46}$ |              $66.39_{\pm0.16}$ |             $\mathbf{78.31}_{\pm0.50}$ |
> | MMActivity |      $59.17_{\pm3.82}$ |          $66.67_{\pm6.29}$ |              $72.50_{\pm2.50}$ |             $\mathbf{75.00}_{\pm6.61}$ |
>
> This performance gap is mechanistically significant: unlike input-level filtering, LIF dynamics are deeply embedded, interacting non-linearly with thresholding and sparsity across all layers. Consequently, an SNN is not functionally equivalent to a filtered ANN. While advanced learnable front-ends are valuable, they introduce optimization variables outside our scope of mechanism-focused comparison. Furthermore, even if a more sophisticated ANN front-end were able to match SNN accuracy, the SNN would still retain its intrinsic low-power advantage.
>
> > **Q2. Regarding Memory Usage on Edge Devices.**
>
> We agree that this point should be clarified more explicitly. During deployment inference, the SNN memory footprint does not scale linearly with $T$: it does not cache membrane potentials for all timesteps, but stores only the current membrane states, requiring little extra memory beyond an ANN. In our additional profiling of SpikingLeNet versus LeNet, with $T=4$, the persistent temporal-state footprint is only 0.636 MB in FP32 (0.318 MB in FP16, estimated). Correspondingly, the total deployment-resident memory is 15.99--16.01 MB for LeNet versus 16.62--16.64 MB for SpikingLeNet, implying an overhead of only $\sim 0.63$ MB. Hence, memory is unlikely to be a major bottleneck on common deployment hardware. For extremely resource-limited devices, this overhead may still matter, but can be further alleviated by reduced precision and memory-compression techniques [2]. Extreme memory optimization is beyond the scope of this work, which focuses on mechanism--data alignment in mmWave SNNs.
>
> > **Q3. From a Theoretical Boundary to a Practical** **$\beta$** **Recommendation.**
>
> We agree that, in practice, a useful method should provide not only a principled boundary but also a concrete recommendation. However, solving for the exact optimum in closed form is generally intractable. Therefore, we do not claim a closed-form exact optimizer. Instead, the practical role of our theory is to shrink the search space. Specifically, we propose a methodological framework, the Selection-via-Pruning procedure: (i) determine the theoretically admissible range, and then (ii) select a concrete $\beta$ from a small discrete candidate set within that range using training data only.
> If a single heuristic recommendation is desired on a coarse discrete grid, a practical rule is to choose the largest candidate not exceeding $\beta^\dagger$. This is consistent with our empirical sweeps: the spectral-matching quantity decreases approximately monotonically with $\beta$, whereas accuracy is clearly non-monotonic. Therefore, the best operating point is expected to occur before the boundary rather than exactly at it. Our current results follow this pattern (e.g., mmFiT: best at $\beta=0.286$ vs. $\beta^\dagger=0.500$; MMActivity: best at $\beta=0.500$ vs. $\beta^\dagger=0.667$). We emphasize that these examples are post-hoc consistency checks, rather than the selection rule itself. We will make this practical interpretation explicit in the revision.
>
> We thank the reviewer again for these constructive insights and will improve figure visibility in the final version.
>
> [1] S. Biswas et al., “PLFNets: Interpretable Complex-Valued Parameterized Learnable Filters for Computationally Efficient RF Classification,” IEEE Trans. Radar Syst., 2024.
>
> [2] Li et al., "Flexnn: Efficient and adaptive dnn inference on memory-constrained edge devices," ACM MobiCom, 2024.

---

> > ### Author Rebuttal · Reviewer_Smez · 2026-04-02
> >
> > All my three comments have been addressed. Especially comment 3, which I was most concerned with, has been answered to an extent. Thus, I am increasing my assesment to 5 points. Thank you for your answers.

---

> > > ### Author Response · Authors · 2026-04-02
> > >
> > > Thank you very much for your thoughtful follow-up and for revisiting our response. We are glad that our clarifications, especially on Comment 3, helped address your main concerns. We sincerely appreciate your updated assessment and constructive feedback.

---

### Official Review · Reviewer_1ECy · 2026-03-10

**Soundness:** 3
**Presentation:** 3
**Significance:** 3
**Originality:** 3
**Overall Recommendation:** 4
**Confidence:** 2

**Summary:**

This paper aims to analyze spiking neural networks in the task of mmWave sensing. Specifically, the authors analyze how the low-pass filtering property interacts with the noisy mmWave signals. Based on the result, the authors propose a frequency-based method to select the membrane decay factor for a given dataset. Experiments show consistent improvement with such designs.

**Compliance With Llm Reviewing Policy:**

Affirmed.

**Final Justification:**

The reviewer addressed the raised question well. However, as I am not very familiar with this area, I tend to maintain my current score.

**Key Questions For Authors:**

1. One question I have, unless I missed it, is whether the $\beta$ selected from the training set can generalize well to the test set. If there is a distribution gap between training and testing data, the optimal $\beta$ may shift. Can the authors conduct additional experiments to evaluate the difference of $\beta$ found on the two sets?

**Limitations:**

yes

**Strengths And Weaknesses:**

I am not very familiar with spiking neural networks, so I may not be able to evaluate this paper as thoroughly or as fairly as an expert in the area. The comments below are therefore based on my current understanding.

### Strengths

1. The idea of leveraging the property of low-pass filtering behavior and further analysis based on the findings is interesting and well-motivated.
2. The paper presents a log of equations and formulas to provide theoretical support.
3. Extensive experiments are conducted showing the effectiveness of their method.

### Weaknesses
1. **Incorrect references.** One issue I noticed is that, while reading the related work for better understanding, I found that some of the references appear to be incorrect. I list them below.

   - Palipana, S., Salami, D., Leiva, L. A., and Sigg, S. V. Proceedings of the ACM on interactive, mobile, wearable and ubiquitous technologies, 5(1):1–27, 2021
     - The paper title is missing.

   - Zhang, M. Artificial intelligence of things: A survey. ACM Transactions on Sensor Networks, 21(1):1–75, 2025.
     - Only the last author is included, while the other authors are missing.

   - Zheng, T., Zhang, Y., Liu, Z., Lin, H., Wang, H., and Luo, J. Neuroradar: A neuromorphic radar sensor for low-power iot systems. In Proceedings of the 21st ACM Conference on Embedded Networked Sensor Systems (SenSys ’23), pp. 223–236, 2023. doi: 10.1145/3625687.3625788.
      - The whole author list appears to be incorrect.

---

> ### Author Rebuttal · Authors · 2026-03-31
>
> Thank you for the thoughtful review and for recognizing the motivation, mechanistic analysis, and empirical consistency of the paper. We especially appreciate your question about whether the $\beta$ selected from training generalizes to held-out test data.
>
> > **Q1. Reference and Formatting Issues.**
>
> We thank the reviewer for identifying these bibliographic and presentation oversights. These are clerical issues only and do not affect the technical content, theoretical derivations, or conclusions. Specifically:
>
> - Ref.1 contains a copy-editing error in which the paper title was inadvertently replaced by the letter “V.”
> - Ref.2 is bibliographically correct; however, in the PDF typesetting, the author list is split across two pages, with all authors except the last appearing at the bottom of page 11 and the final author together with the remaining citation information appearing at the top of page 12, which may hinder readability.
> - Ref.3 contains an author-list mismatch introduced during manual pasting and editing.
>
> In the final version, we will conduct a comprehensive audit of the entire bibliography to ensure all references are accurate, complete, and professionally formatted.
>
> > **Q2.**  **Generalization of the Selected** $\beta$ **.**
>
> Thank you for this important question. Following the reviewer’s suggestion, for this additional analysis we use a 6:2:2 train/validation/test split, and select the recommended $\beta$ by scanning the theory-derived candidate set on the validation set only; the test set is reserved strictly for final evaluation.
>
> To assess whether this validation-selected $\beta$ generalizes to held-out data, we conducted an additional post-hoc diagnostic analysis on Pantomime using SpikingLeNet+LIF over three random seeds. Specifically, we compared the validation-selected $\beta$ with a test-oracle $\beta$ obtained by scanning the same candidate set on the test set for analysis only, not for model selection. The original train/validation/test split and training pipeline were kept unchanged. Moreover, the additional perturbations were introduced only post hoc on copies of the held-out test set, and were not used for training, validation, or model selection. We evaluated the clean test set together with several mild, label-preserving shifts: GaussianNoise (20% additive noise), GainScale ($1.20\times$ global scaling), TemporalDrop (dropping one frame), TemporalShift ($\pm 1$ frame offset), and PointDropout (dropping 10% of input points).
>
> As shown below, the validation-selected $\beta$ remains close to the test-oracle $\beta$, with exact matches in many seeds and only small accuracy differences even when the oracle shifts. Overall, these results suggest that the selected $\beta$ generalizes reasonably well to held-out data and remains fairly robust under mild test-time distribution shift.
>
> | Dataset   | Condition     | Severity | Train/val $\beta$            | Test $\beta$                 | Acc@Train-val $\beta$ (%)     | Acc@Test $\beta$ (%)          |
> | ----------- | --------------- | ---------- | ----------------------- | ----------------------- | ----------------------- | ----------------------- |
> | Pantomime | Clean         | -        | 0.259 / 0.231 / 0.333 | 0.259 / 0.259 / 0.333 | 79.72 / 76.20 / 77.46 | 79.72 / 77.75 / 77.46 |
> | Pantomime | GaussianNoise | 20%      | 0.259 / 0.231 / 0.333 | 0.259 / 0.412 / 0.375 | 77.18 / 74.65 / 71.27 | 77.18 / 75.35 / 74.65 |
> | Pantomime | GainScale     | 1.20x    | 0.259 / 0.231 / 0.333 | 0.259 / 0.259 / 0.333 | 78.73 / 77.04 / 77.61 | 78.73 / 78.59 / 77.61 |
> | Pantomime | TemporalDrop  | 1 frame  | 0.259 / 0.231 / 0.333 | 0.259 / 0.231 / 0.333 | 74.08 / 74.08 / 72.39 | 74.08 / 74.08 / 72.39 |
> | Pantomime | TemporalShift | $\pm$1 frame  | 0.259 / 0.231 / 0.333 | 0.259 / 0.231 / 0.333 | 74.65 / 71.41 / 71.41 | 74.65 / 71.41 / 71.41 |
> | Pantomime | PointDropout  | 10%      | 0.259 / 0.231 / 0.333 | 0.333 / 0.429 / 0.535 | 68.73 / 65.77 / 65.77 | 69.58 / 66.06 / 68.59 |
>
> We also observe that under stronger structured shifts (e.g., PointDropout with a 30% drop ratio), the test-oracle $\beta$ can deviate more substantially from the training-selected one. This is consistent with the reviewer’s concern that a larger train--test distribution gap may shift the optimal $\beta$. These results validate that our principled selection is effective for deployment, while also motivating future research into online/test-time adaptation of $\beta$ for highly non-stationary environments.
>
> Thank you again for the careful reading and helpful suggestions.

---

> > ### Author Rebuttal · Reviewer_1ECy · 2026-04-02
> >
> > The authors have addressed my concern. However, as I am not familiar with this area, I tend to maintain my current score.

---

> > > ### Author Response · Authors · 2026-04-02
> > >
> > > We sincerely appreciate your consideration of our response. We are grateful that our response addressed your concern, and we fully respect your decision to maintain the current score. Thank you again for your time and evaluation.

---

### Official Review · Reviewer_SWdq · 2026-03-12

**Soundness:** 3
**Presentation:** 2
**Significance:** 2
**Originality:** 3
**Overall Recommendation:** 4
**Confidence:** 3

**Summary:**

This paper studies spiking neural networks (SNNs) for mmWave sensing from a mechanism–data alignment perspective, deriving a principled criterion to configure the membrane decay factor of leaky integrate-and-fire (LIF) neurons by matching their effective bandwidth to the data’s discriminative spectral content.

**Compliance With Llm Reviewing Policy:**

Affirmed.

**Final Justification:**

I acknowledge the excellent performance of the SNN-mmWave integration, and the authors have presented a compelling narrative in the paper. That said, this work is essentially a simplistic "A+B" combination, with only minor modifications to the spiking neuron model. Furthermore, the authors only conduct Matched Family Fairness experiments by reducing the training data volume on small-scale datasets to demonstrate the superiority of SNNs over ANNs.
**My assessment leans toward the borderline, yet this recommendation option is not available**. I argue that the authors should design task-specific SNN network architectures tailored for mmWave sensing, instead of rigidly applying the SNN paradigm to generic off-the-shelf ANN architectures.

**Key Questions For Authors:**

1. I am curious about the respective training and test accuracies of the compared methods. According to my reasoning, most ANN methods can achieve nearly 100% training accuracy but perform poorly on the test set.

2. I am curious about the parameter settings of MLP, RNN, GRU, Bi-LSTM, and ViT.

**Strengths And Weaknesses:**

Strength

1. Deep mechanistic insight via mechanism-data alignment, offering interpretable LIF parameter tuning guidelines.

2. Dual advantages of 6.22% higher accuracy and 3.64× lower energy consumption vs. ANN baselines.

3. The authors conducted sufficient ablation experiments and visualizations to validate the frequency-matching hypothesis and method efficacy.

Weakness

1. The authors did not make innovations in the network architecture; they merely applied SNNs to the mmWave domain and provided guidance on the setting of neuron hyperparameters.

2. The authors' writing lacks professionalism, particularly in the Introduction section, and it is recommended to refer to published papers for improvement.

3. The compared methods are general-purpose rather than mmWave-specialized, which tends to cause overfitting on small mmWave datasets. Although the authors propose a reasonable mechanism story, SNN’s timesteps naturally capture temporal dynamics, making this comparison unfair from an engineering viewpoint.

---

> ### Author Rebuttal · Authors · 2026-03-31
>
> Thank you for recognizing our mechanistic insight and practical value, and for raising concerns about novelty, presentation, and fairness.
>
> > **Q1. Theoretical and Methodological Contribution Beyond Architectural Novelty.**
>
> We clarify that our work is not a straightforward application of SNNs to mmWave sensing, but a mechanism-driven study with rigorous theoretical support. Beyond empirical tuning, we provide a mechanistic analysis that explains why spiking dynamics are well matched to the spectral structure of mmWave signals. Crucially, we derive a principled criterion for the membrane decay factor, transitioning from heuristic settings to a theoretically grounded methodology. As demonstrated in Appendix G, these benefits generalize across diverse backbones (e.g., SEW-ResNet), proving that our contribution reflects a general property of SNN dynamics rather than an architecture-specific artifact.
>
> > **Q2. Fairness of the Comparison.**
>
> To isolate neuron dynamics, we use a unified protocol with general-purpose baselines. Our goal is not to outperform every mmWave-specific ANN by architectural design, but to test whether LIF dynamics provide a stronger inductive bias for mmWave sensing. Adding domain-specific modules would confound architectural gains with the proposed mechanism. This is also not a temporal-versus-non-temporal comparison, since some baselines already model time explicitly (e.g., LSTM). The performance gap suggests that LIF dynamics offer a more effective structural match for mmWave signals than standard recurrent units.
>
> To further verify this, we additionally compare against the recent mmWave-specific model channel-DN4 [1] over three random seeds. As shown below, SpikingLeNet remains consistently more accurate across all four datasets. These results suggest that the observed advantage is not merely an artifact of using only generic baselines, but reflects a fundamental mechanism-data alignment that holds even against specialized architectures.
>
> | **Model**            | **AOPHand** | **mmFiT** | **Pantomime** | **MMActivity** |
> | ------------- | -- | -- | -- | -- |
> | channel-DN4 | $81.05_{\pm1.54}$ | $68.28_{\pm0.52}$ | $59.72_{\pm1.93}$ | $55.00_{\pm5.00}$ |
> | **SpikingLeNet**            | $\mathbf{83.70}_{\pm4.24}$ | $\mathbf{73.67}_{\pm1.55}$ | $\mathbf{78.31}_{\pm0.50}$ | $\mathbf{75.00}_{\pm6.61}$ |
>
> > **Q3. Train/Test Accuracies.**
>
> Using the same split and training protocol as in the submission (150 epochs, Adam, initial LR 1e-3, cosine schedule, batch size 64, weight decay 0), we report both the final-epoch train/test accuracies and the train/test accuracies at the peak test epoch on AOPHand a representative seed.
>
> | Model     | Final (Tr/Test) | Peak-test (Tr/Test) |
> | ----------- | ----------------: | --------------------: |
> | LeNet     |    100.00/62.18 |         99.40/62.57 |
> | MLP       |    100.00/65.74 |         82.69/70.69 |
> | RNN       |     17.96/21.98 |         17.66/21.98 |
> | Bi-LSTM   |     25.25/23.56 |         25.00/23.76 |
> | CNN-GRU   |    100.00/65.35 |         87.45/67.13 |
> | GRU       |     62.75/60.20 |         61.61/60.99 |
> | LSTM      |     69.89/64.55 |         69.20/65.15 |
> | ResNet18  |     99.95/68.32 |         82.79/71.09 |
> | ResNet50  |    100.00/68.51 |         75.55/72.67 |
> | ResNet101 |     98.96/65.94 |         94.25/69.50 |
> | VGG9      |    100.00/71.49 |         73.02/74.26 |
> | VGG16     |     62.55/48.32 |         53.37/60.99 |
> | ViT       |     17.96/21.98 |         16.37/21.98 |
>
> These results do not support a single ANN overfitting story, but instead point to heterogeneous model--data mismatch and optimization behavior. This is why we emphasize mechanism--data alignment rather than a purely overfitting-based explanation.
>
> > **Q4. Parameter Settings.**
>
> For clarity, the main settings are: MLP: two hidden layers (1024, 128) with ReLU; RNN/GRU/LSTM: hidden size 128, one recurrent layer, sequence length 16, classification from the final hidden state; Bi-LSTM: hidden size 128, one bidirectional layer; ViT: input 64×64, patch size 16×16, embedding dim 256, depth 4, 4 heads. These settings are standard and broadly consistent with prior sensing benchmarks such as SenseFi (Appendix E). For reproducibility, we detail the unified training protocol in Appendix E and provide the full implementation in the supplementary code.
>
> > **Q5. Gain Not Due to SNNs Per Se.**
>
> SNN performance gains are not inherent but contingent on mechanism-data alignment. On AOPHand, SpikingLeNet achieves 83.70% accuracy under the matched setting $\beta=0.500$, whereas mismatching $\beta=0.875$ degrades performance to 73.53%. This validates our core contribution: transforming empirical tuning into a principled alignment methodology.
>
> Thank you again for these helpful suggestions. We will revise the Introduction and improve the writing and presentation.
>
> [1] Fan et al., "Few-shot Human Motion Recognition through Multi-Aspect mmWave FMCW Radar Data," IGARSS 2025.

---

> > ### Author Rebuttal · Reviewer_SWdq · 2026-04-04
> >
> > Thank you for the clarification and for providing additional train/test accuracies and implementation details. I appreciate the authors’ effort to address my questions.
> >
> > That said, I remain unconvinced that the current evidence is sufficient to support the paper’s central mechanistic claim. In my view, a more parsimonious explanation for the observed advantage of SNNs is improved generalization due to stronger inductive bias or implicit regularization, especially given the relatively small scale of the mmWave sensing datasets considered here. The additional train/test results suggest that several ANN baselines suffer from substantial train-test gaps, which makes overfitting or model-data mismatch a plausible alternative explanation.
> >
> > Therefore, the key issue is not whether the proposed frequency-matching story is intuitively appealing, but whether the current experiments adequately distinguish this explanation from a simpler regularization/generalization account. At present, I do not believe the rebuttal fully rules out the latter. In particular, the current evidence does not yet make it clear whether the gains should primarily be attributed to the proposed mechanism, or instead to the fact that SNNs impose a stronger structural bias that happens to generalize better on small and noisy datasets.
> >
> > To make the mechanistic claim more convincing, I would have liked to see stronger controls addressing several questions. For example:
> > (1) whether the comparison remains in favor of SNNs after carefully matching model capacity and tuning regularization strength for ANN baselines;
> > (2) whether the advantage persists under different levels of data scarcity, which would help determine whether the gain is mainly a generalization effect;
> > (3) whether comparable improvements can be obtained by equipping ANN baselines with simple frequency-aware or low-pass filtering modules, which would test whether the claimed benefit is truly specific to spiking dynamics;
> > (4) whether the best membrane decay factor consistently tracks changes in the data spectrum across datasets or settings, rather than simply acting as a task-dependent regularization knob;
> >
> > More broadly, I am not yet persuaded that SNNs should significantly outperform well-tuned ANNs on this task for the reasons claimed here. Thus, while I acknowledge the paper’s interesting perspective and the authors’ effort in the rebuttal, my main concern about the interpretation of the gains remains unresolved.

---

> > > ### Author Response · Authors · 2026-04-07
> > >
> > > We thank the reviewer for raising this important alternative explanation. We agree that stronger inductive bias and improved generalization may contribute. Our claim is narrower: the benefit of LIF dynamics is not fully explained by generic regularization alone. Instead, the observed behavior shows systematic dependence on the discriminative spectrum and supports a principled criterion for selecting the membrane decay factor.
> > >
> > > > **Q1.**  **Matched Family Fairness.**
> > >
> > > We agree that a key control is whether the SNN--ANN gap persists under matched backbone family, capacity, tuning opportunity, and model-selection protocol. On AOPHand, we built a parameter-matched non-spiking LeNet from the same macro-architecture and evaluated it against the SNN under the same split, selection rule, training budget, and hyperparameter search (14 trials per seed per fraction). [1] Under this controlled setting, the SNN still substantially outperforms the matched ANN across all training fractions (3 seeds).
> > >
> > > | Train Fraction | ANN Test Acc. | SNN Test Acc. | Gap | $\beta^*$ |
> > > | ---------------- | --------------- | --------------- | ----- | -- |
> > > | 100%           | $54.85_{\pm1.54}$              | $85.61_{\pm1.45}$              | $+30.76$    | $0.188_{\pm0.030}$ |
> > > | 50%            | $44.69_{\pm1.21}$              | $78.55_{\pm2.43}$              | $+33.86$    | $0.181_{\pm0.080}$ |
> > > | 25%            | $37.36_{\pm2.18}$              | $69.04_{\pm2.88}$              | $+31.68$    | $0.202_{\pm0.082}$ |
> > >
> > > This matched-family control is a fairness test rather than an ANN upper bound: it does not claim that no ANN can close the gap, but it does show that the observed SNN advantage is not due to mismatched backbone family, parameter count, tuning opportunity, or model-selection protocol.
> > >
> > > > **Q2. Data Scarcity.**
> > >
> > > We agree that a small-data generalization account is plausible. However, our data-scarcity sweep does not support it as the main explanation. Under the matched setting in Q1, the SNN retains similarly large gains across training fractions, while neither the selected $\beta^*$ values (0.188/0.181/0.202) nor the subset-level spectral summaries shows the monotonic stronger-smoothing trend this account would predict. Nor is there a low-frequency shift; if anything, the spectral centroid increases slightly from 0.65 to 0.69.
> > >
> > > We therefore conclude conservatively that improved generalization may contribute, but is not sufficient to explain the full observed pattern.
> > >
> > > > **Q3. ANN-side Filtering Controls.**
> > >
> > > We agree that a natural control is to test whether simple ANN-side low-pass filtering can reproduce the SNN gain. As shown in Fig. 5 of the main paper, across all tested cutoffs (0.1–0.5 × Nyquist) and over 3 random seeds, fixed LPF preprocessing improves LeNet; however, it still remains well below SpikingLeNet.
> > >
> > > | Dataset    | LeNet | Best cutoff | Best LeNet+LPF | SpikingLeNet |
> > > | ------------ | ------- | ------------- | ---------------- | -------------- |
> > > | Pantomime  | $61.83_{\pm0.42}$      | 0.4         | $65.49_{\pm2.46}$               | $\mathbf{78.31}_{\pm0.50}$             |
> > > | MMActivity | $59.17_{\pm3.82}$      | 0.3         | $66.67_{\pm6.29}$               | $\mathbf{75.00}_{\pm6.61}$             |
> > >
> > > We therefore interpret this control conservatively: ANN-side filtering is clearly helpful, which supports the relevance of frequency-selective processing, but the current ANN filtering controls do not fully reproduce the SNN gain.
> > >
> > > > **Q4.**  **Controlled Spectrum–β Co-Shift.**
> > >
> > > We further tested whether the preferred membrane decay shows spectrum-dependent behavior beyond a purely task-level regularization interpretation. Across datasets, the dataset-level spectrum-$\beta$ correspondence is already shown in Fig. 4 of the main paper: datasets with relatively more high-frequency discriminative content prefer smaller $\beta$, while those with relatively more low-frequency content prefer larger $\beta$. More importantly, we also tested a controlled same-task shift. On mmFiT, stronger temporal smoothing of the same task (long setting) shifts discriminative content toward lower frequencies, evidenced by a lower $\omega_{80}/\pi$ (the normalized 80%-energy frequency) and both $\beta^\dagger$ and $\beta^*$ increase accordingly.
> > >
> > > Thus, the preferred $\beta$ tracks spectral variation not only across datasets but also under a controlled within-task manipulation.
> > >
> > > | Setting | $\omega_{80}/\pi$ | $\beta^\dagger$ | $\beta^*$ |
> > > | --------- | -- | -- | -- |
> > > | base    | $0.844_{\pm0.003}$ | $0.583$ | $0.448_{\pm0.074}$ |
> > > | long    | $0.594_{\pm0.002}$ | $0.615$ | $0.557_{\pm0.074}$ |
> > >
> > > Thank you again for this helpful comment. We interpret the evidence conservatively: while generic inductive-bias effects may contribute, the overall pattern is more consistent with $\beta$ tracking the discriminative spectrum than with generic regularization alone.
> > >
> > > [1] Bouthillier et al. "Accounting for variance in machine learning benchmarks." MLSys 2021.

---

### Decision · Program_Chairs · 2026-04-30

**Decision:**

Accept (regular)

**Comment:**

This paper provides a principled mechanism-data alignment framework for SNNs in mmWave sensing. Unlike empirical tuning, it derives a frequency-matching criterion for the membrane decay factor, transforming heuristic hyperparameter selection into a theoretically grounded methodology. Extensive experiments across four datasets show consistent accuracy gains (6.22% average) and 3.64× energy reduction over ANNs. The rebuttal convincingly addresses concerns about fairness, generalization, and memory overhead, including matched-family controls, data-scarcity sweeps, and ANN-side filtering comparisons. Three reviewers recommend accept/weak accept, with one raising the score to 5 after rebuttal. The work offers strong theoretical novelty, empirical rigor, and practical impact for edge perception.